# Chimeric inheritance and crown-group acquisitions of carbon fixation genes within Chlorobiales: Origins of autotrophy in Chlorobiales and implication for geological biomarkers

**Madeline M. Paoletti** [1,2], **Gregory P. Fournier** [1]*

**1** Department of Earth, Atmospheric, & Planetary Sciences, Massachusetts Institute of Technology, Cambridge, MA, United States of America, **2** Department of Biological Sciences, Wellesley College, Wellesley, MA, United States of America

* g4nier@mit.edu

**Data Availability Statement:** All data can be found in the Dryad Data Repository (https://datadryad.org/stash), under doi:10.5061/dryad.931zcrjmw.

## Abstract

The geological record of microbial metabolisms and ecologies primarily consists of stable isotope fractionations and the diagenetic products of biogenic lipids. Carotenoid lipid biomarkers are particularly useful proxies for reconstructing this record, providing information on microbial phototroph primary productivity, redox couples, and oxygenation. The biomarkers okenane, chlorobactane, and isorenieratene are generally considered to be evidence of anoxygenic phototrophs, and provide a record that extends to 1.64 Ga. The utility of the carotenoid biomarker record may be enhanced by examining the carbon isotopic ratios in these products, which are diagnostic for specific pathways of biological carbon fixation found today within different microbial groups. However, this joint inference assumes that microbes have conserved these pathways across the duration of the preserved biomarker record. Testing this hypothesis, we performed phylogenetic analyses of the enzymes constituting the reductive tricarboxylic acid (rTCA) cycle in Chlorobiales, the group of anoxygenic phototrophic bacteria usually implicated in the deposition of chlorobactane and isorenieretane. We find phylogenetically incongruent patterns of inheritance across all enzymes, indicative of horizontal gene transfers to both stem and crown Chlorobiales from multiple potential donor lineages. This indicates that a complete rTCA cycle was independently acquired at least twice within Chlorobiales and was not present in the last common ancestor. When combined with recent molecular clock analyses, these results predict that the Mesoproterozoic lipid biomarker record diagnostic for Chlorobiales should not preserve isotopic fractionations indicative of a full rTCA cycle. Furthermore, we conclude that coupling isotopic and biomarker records is insufficient for reliably reconstructing microbial paleoecologies in the absence of a complementary and consistent phylogenomic narrative.

**Funding:** This work was supported by the National Science Foundation Integrated Earth Systems award EAR grant no. 1615426 to GPF (https://www.nsf.gov/pubs/2016/nsf16589/nsf16589.htm). The funders had no role in study design, data collection and analysis, decision to publish, or preparation of the manuscript.

**Competing interests:** The authors have declared that no competing interests exist.

## Introduction

Lipid biomarkers are geochemically stable molecular remnants of organisms, frequently used to trace paleoenvironmental proxies, or as "fossils" indicating the presence of specific groups of organisms [1–3]. Phototrophic bacteria, including oxygenic cyanobacteria and anoxygenic green sulfur bacteria (GSB, order Chlorobiales) and purple sulfur bacteria (PSB, order Chromatiales), have especially significant biomarker records in the form of carotenoid pigment derivatives. These can indicate periods of major planetary change, such as great oceanic anoxic events [4].

The first appearance of sulfur bacteria in the biomarker record is from the 1.64 Ga Barney Creek Formation, which contains carotenoids including okenane, chlorobactane, and isorenieratane [5, 6]. This implies sulfur bacteria were a substantial part of aquatic microbial ecologies by this time, consistent with the presence of a stratified ocean with widespread euxinic conditions [7]. However, the genes encoding the biosynthesis of aromatic carotenoids and their derivatives have also been identified in cyanobacteria, which occupy more oxygenated water columns [8–10]. Carotenoid biomarkers within Neoproterozoic marine sediments (1000–542 Ma) have shown to be primarily comprised of cyanobacterial renierapurpurane, with small amounts of the Chlorobiaceae-associated biomarker isorenieratane. Furthermore, cyanobacteria producing synechoxanthin have been shown to be capable of producing both isorenieratene and, in principle, chlorobactene, given their enzymatic repertoire [11]. This brings the evidence for GSB at 1.64 Ga BCF formations into question, as well as the practice of using these biomarkers as a calibration for dating Chlorobiales in molecular clock analyses, since cyanobacteria could potentially be the source of the carotenoid derivatives found in these deposits [12, 13].

A distinct, comparable source of information for tracing microbial autotrophy can be found within the geochemical record of carbon fixation, which can further inform reconstructions of microbial paleoecology. There are six known modern biological pathways used by organisms to fix $CO_2$. Most fractionate carbon differently, variably offsetting carbon isotope ($\delta$13C) values of inorganic and organic carbon in the isotopic record [14, 15]. Through this bulk signaling of $\delta$13C sources, a cycle's input can be mathematically inferred by isotopic mass balance of its components [16].

Of the known carbon fixation pathways, the Calvin-Benson-Bassham (CBB) cycle and the rTCA cycle are the most prevalent. The taxonomic distributions of these pathways are polyphyletic, consistent with deep evolutionary histories of horizontal gene transfer(HGT) [16, 17]. The CBB cycle is found in both photoautotrophic bacteria such as cyanobacteria and purple sulfur bacteria, as well as some chemolithoautotrophs; it fixes three molecules of $CO_2$ for the synthesis of one 3-phoshoglycerate molecule using ribulose-1,5-bisphosphate carboxylase/oxygenase (RuBisCo). The rTCA cycle is present in a diverse group of autotrophic microbes, including most members of Chlorobiales, sulfate-reducing bacteria, hyperthermophilic bacteria, and Crenarchaeota [18, 19]. The complete cycle fixes three molecules of $CO_2$ for the synthesis of one 3-carbon pyruvate molecule through 9 steps that also produce several metabolic intermediates [20–22] (Fig 1). Except for ATP-citrate lyase, which drives the reductive cycling [23], and pyruvate: ferredoxin oxioreductase and 2-oxoglutarate:ferredoxin oxioreductase (OGOR) needed to reduce $CO_2$, these enzymes are also used within the oxidative TCA cycle, which has been shown to operate within Chlorobiales under some growth conditions [24]. However, a complete oxidative TCA cycle is not present in obligate anaerobes, as aerobic respiratory chains are presumably necessary for the efficient re-oxidation of NADH [25].

Recently, an alternative rTCA cycle has been characterized that uses two different enzymes to convert citrate into citryl-CoA, instead of ATP citrate lyase [26]. This was initially observed

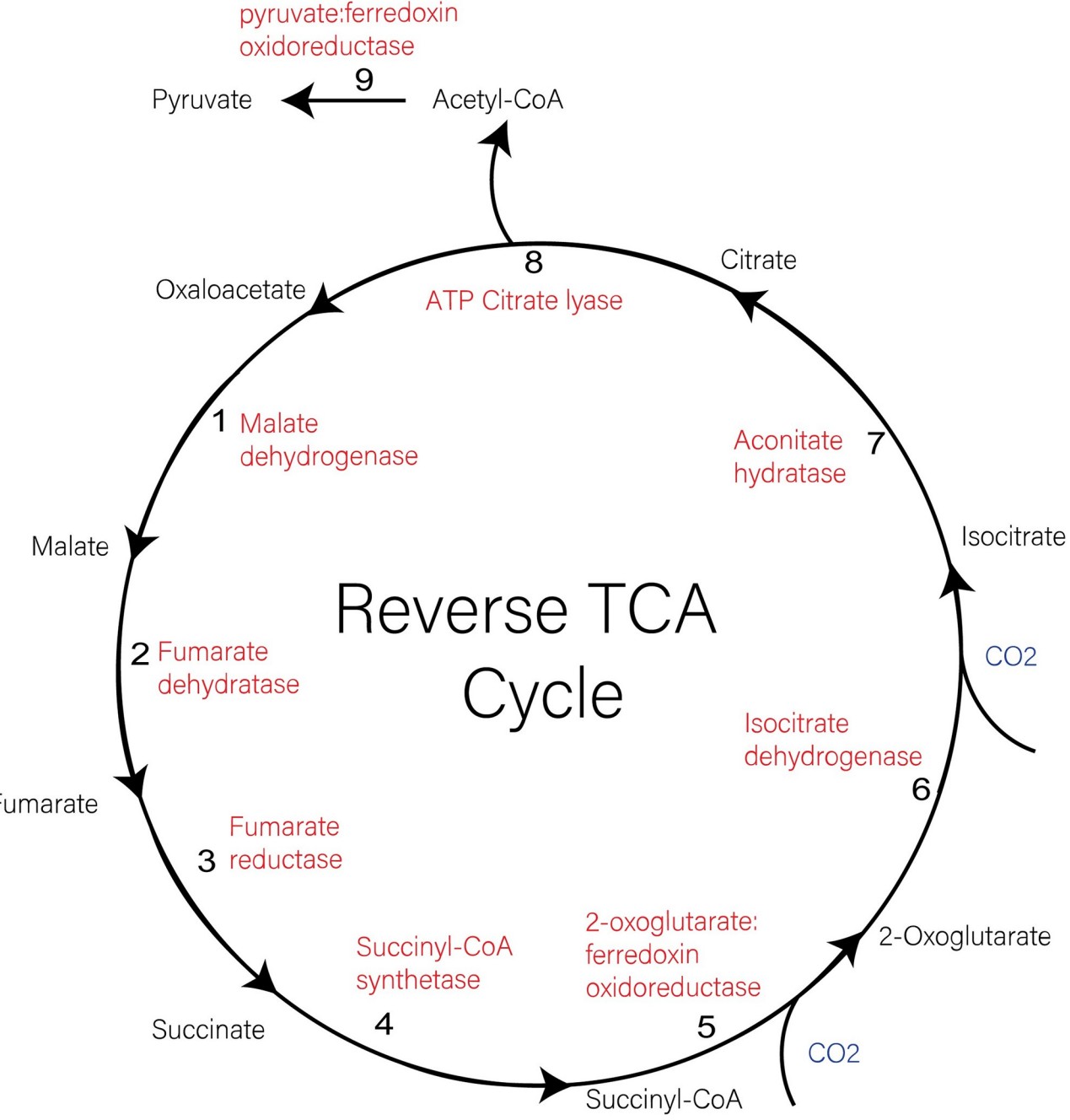

**Fig 1. Enzymatic map of the rTCA cycle.** Enzymes catalyzing each reaction (red), inputs of carbon dioxide (blue) and intermediate compounds (black) are shown. Steps are numbered in the direction of reactions, starting with the conversion of oxaloacetate to malate via malate dehydrogenase.

in *Hydrogenobacter thermophilus* but is also present in other members of Aquificeae [27]. Anaerobes with partial TCA cycles also use α-ketoglutarate:ferredoxin oxioreductase, and fumarate reductase instead of succinate dehydrogenase, similar to the rTCA cycle [25, 28], and pyruvate: ferredoxin oxioreductase is used in some carbohydrate fermentation pathways [29]. Thus, the phyletic distribution of genes found in the rTCA cycle is complex, and potentially indicative of several different metabolic processes other than autotrophy.

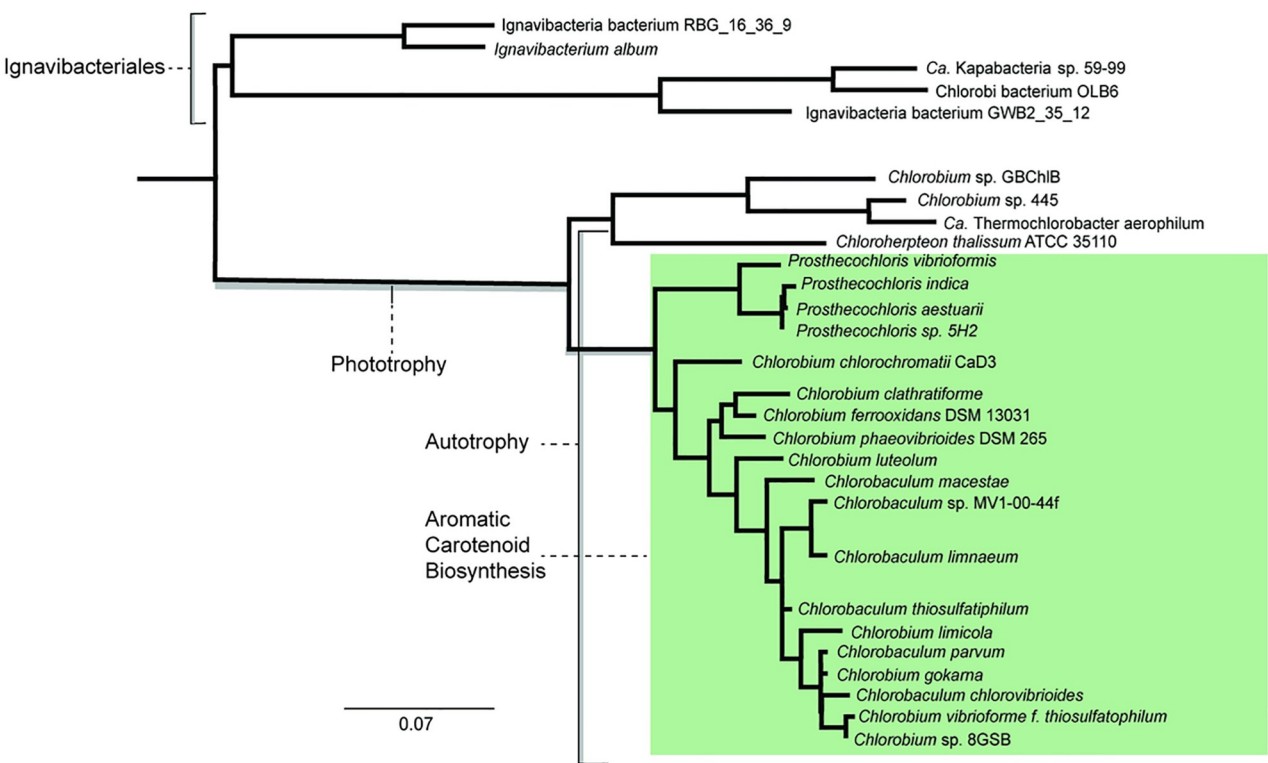

**Fig 2. 16S rRNA tree of Chlorobiales and Ignavibacterium (outgroup) showing taxonomic distributions of a selection of traits.** Labeled brackets indicate the taxonomic distribution of phototrophy and autotrophy within this group. The green box indicates groups known to synthesize aromatic carotenoid lipid biomarkers.

The order Chlorobiales consists of a group of unclassified photoheterotrophic thermophiles and two families: Chlorobiaceae that contains five genera (*Chlorobium, Chlorobaculum, Chloroherpeton,* and *Prosthecochloris*) and the newly proposed family Chloroherpetonaceae with species of *Chloroherpeton* [30, 31]. The 16S rRNA phylogenies of this group recover close relationships to Ignavibacterales (within the phylum Chlorobi) and the Bacteriodetes, which are often grouped together along with Firmicutes in a CFB superphylum [32, 33] (Fig 2). With the exception of the unclassified thermophiles, Chlorobiales are strictly anaerobic, obligate photoautotrophs that use the rTCA cycle. Within this group, *Chlorobium, Chlorobaculum,* and *Prosthecochloris* genera form a clade that synthesizes aromatic carotenoid compounds. Metagenomic analyses and culturing of novel strains has revealed more aerobic and photoheterotrophic members of Chlorobi lacking enzymes for autotrophic carbon fixation [34–37]. As shown in Fig 2, some of these uncultured genome species, such as *Candidatus Thermochlorobacter aerophylum* [34, 35], *Chlorobium* sp. 445 and *Chlorobium* sp. GBChlB35 group as a distinct clade within Chlorobiales. This distribution of physiological traits in Chlorobiales may be explained by vertical inheritance and loss, or by HGT, depending on the inferred phylogenies of the individual gene families providing these functions.

If the constituent enzymes of the rTCA cycle were acquired within crown Chlorobiales, rather than the stem lineage, then ancestral autotrophy cannot be inferred for this order. Subsequently, diagnostic lipid biomarkers more ancient than the inferred age of crown Chlorobiales should not show isotopic fractionations indicative of rTCA cycle carbon fixation. Conversely, a phylogenetic signal showing clear acquisition of all rTCA enyzmes within stem

**Table 1. Summary of evolutionary histories of carbon fixation genes in Chlorobiales.**

| rTCA Step | Enzyme Name | Inferred History | Sibling Groups | Outgroup Carbon Metabolism |
|---|---|---|---|---|
| 1 | malate dehydrogenase | Chlorobiales: stem HGT | *Rhodothermus marinus* | Mixed |
| 2 | fumarate hydratase | (A)Chlorobiales: stem HGT/vertical inheritance (B) *C.thalissium*: orthologous displacement | (A) Bacteroidetes, Ignavibacterium, Plantomycetaeceae (B) *Oceanospiralles* bacterium | Heterotrophs |
| 3 | fumarate reductase | Chlorobiales: stem HGT | Candidatus Lambdaproteobacteria, *Geobacter*, Desulfuromonadales bacterium | Mixed |
| 4 | succinyl-CoA synthetase | Chlorobiales: vertical inheritance | Bacteroidetes, Ignavibacterium | Heterotrophs |
| 5 | 2oxoglutarate: ferredoxin oxidoreductase | Chlorobiales: stem HGT from Bacteroidetes | Balneolaceae, Bacteroidetes | Heterotrophs |
| 6 | isocitrate dehydrogenase | (A)Chlorobiales: HGT (B)*C.thalissium*/ Photoheterotrophs: HGT | (A) Deltaproteobacteria, Nitrospira/ Nitrospinae (B) Armatimonadetes, Caldithrix | Mixed |
| 7 | aconitate hydratase | (A)Chlorobiales: HGT (B)*C.thalassium*/ Thermochlorobacter: HGT | (A) Campylobacterales (B)Ignavibacteria, Spirochaetes | (A) Autotrophs (B) Heterotrophs |
| 8 | ATP citrate lyase | Chlorobiales: stem HGT | Nitrospirae/ Nitrospinae | Mixed |
| 9 | pyruvate: erredoxin oxidoreductase | Chlorobiales: HGT | Bacteroidetes, Ignavibacteria | Mixed |

Table notes: Italicized species names refer to taxon source of query sequence for respective groupings

Chlorobiales would be supportive of a much older history of autotrophy within this group and would predict isotopic fractionations of even the oldest preserved biomarkers to be similar to those of modern autotrophic Chlorobiales. To investigate this, we performed phylogenetic analyses of rTCA cycle enzymes in Chlorobiales to map their evolutionary histories. Overall, our investigations reveal a mixed history of vertically inherited and horizontally transferred rTCA cycle genes across multiple groups of Chlorobiales ([Table 1]). These complex histories support a hypothesis of patchwork enzymatic inheritances from multiple and diverse HGT donors, implying the complete rTCA cycle, and therefore autotrophy, was not present in the ancestral stem group of Chlorobiales.

## Materials and methods

### Genomic data retrieval

Query sequences for rTCA cycle enzymes were selected from the sequenced *Chlorobium thalissium* ATCC 35110 genome (CP001100.1) retrieved from the NCBI database ([S1 Table]). In cases where the protein ortholog from *Chloroherpeton thalissum* did not recover monophyly with other Chlorobiaceae, orthologous query sequences were taken from *Chlorobium tepidum* TLS (AE006470.1).

### BLAST search for homologous enzymes

The Basic Local Alignment Search Tool (BLAST) was used to compare queries with homologous sequences using default search parameters. The BLASTp cutoff for homologous identification was set at 250 taxa maximum, with sequence identity $\geq$ 32% These searchers were not exhaustive for all rTCA homologs, but rather exhaustively identified all Chlorobiales homologs and extensively sampled sequences representing potential HGT donor lineages and/or distant vertically inherited sequences from outgroup clades.

### Phylogenetic analysis

Protein sequences were aligned using the multiple sequence alignment tool MAFFT v.7.245 (FFT-NS-2, BLOSUM6) [38]. Maximum-likelihood phylogenetic trees were generated using IQTree run with the Bayes Information Criterion (BIC) test (S1 Table). Support for bipartitions was inferred using rapid bootstraps (1000 replicates) and SH-aLRT tests. Additional phylogenetic analysis of the ATP citrate lyase dataset was performed using Phylobayes v. 4.171, using C20 site specific profiles and a convergence criterion cutoff of 0.3 for all parameters, with a 20% chain burn-in. Trees were rooted at midpoints and visualized in FigTree v.1.4.4.

## Results

### rTCA cycle protein phylogenies show a complex history of HGT

Maximum-likelihood phylogenies of rTCA pathway proteins within Chlorobiales show surprisingly diverse evolutionary origins (Table 1). Several show monophyly indicative of acquisition within stem Chlorobiales (malate dehydrogenase, pyruvate ferredoxin, ATP citrate lyase, succinyl-CoA synthetase, OGOR, fumarate reductase). Other enzymes show polyphyletic distributions, indicative of multiple HGT acquisitions from different donor lineages (isocitrate dehydrogenase, fumarate hydratase, aconitate hydratase). Several enzymes were subsequently lost within the photoheterotrophic Chlorobiales clade (aconitate hydratase, pyruvate: ferredoxin oxidoreductase, fumarate reductase, ATP-citrate lyase), likely following divergence from the *Chloroherpeton* lineage.

Several general trends can be identified in the placement of these groups within broader protein diversity, based on the outgroup sequences within each tree. There are three scenarios wherein the inheritance of an rTCA cycle protein could be inferred for crown Chlorobiales. First, trees where the outgroups are similar to those of the species tree, such as other Chlorobi (e.g., Ignavibacteria and/or Bacteroidetes) suggest that these proteins were ancestral within the Chlorobiales lineage, and subsequently vertically inherited. Of the rTCA pathway proteins, only succinyl-CoA synthetase appears to have a history consistent with this interpretation. Second, trees where Chlorobiales sequences are on long branches without any closely related outgroups and crown group monophyly is preserved (malate dehydrogenase, pyruvate:ferredoxin, ATP citrate lyase). These enzymes were likely acquired by stem Chlorobiales via HGT, but current taxonomic sampling and/or lack of phylogenetic signal prevent an unambiguous identification of a putative donor lineage. Third, trees showing vertical inheritance within crown Chlorobiales following an HGT from a clear donor group to stem Chlorobiales. Only OGOR preserves this signal. The remaining tree topologies are complicated by HGTs within crown group lineages, so that Chlorobiales is polyphyletic within the tree (fumarate hydratase, isocitrate dehydrogenase, aconitate hydratase). Either these genes were acquired in stem Chlorobiales and a subsequent orthologous displacements occurred, or the original acquisitions within crown Chlorobiales were independent.

**Malate dehydrogenase.** The ML tree for malate dehydrogenase shows a broad taxonomic range of bacteria, with 11 distinct phyla represented in 242 taxa. The tree is consistent with acquisition of malate dehydrogenase in the Chlorobiales stem ancestor, and subsequent vertical inheritance. The sibling groups are diverse, including both autotrophic and heterotrophic representatives of Gemmatimonadetes, Deltaproteobacteria, Acidobacteria, and Bacteroidetes, with poorly supported relationships between most groups (Fig 3). This indicates a complex history of HGT between phyla, preventing identification of a likely HGT donor to stem Chlorobiales.

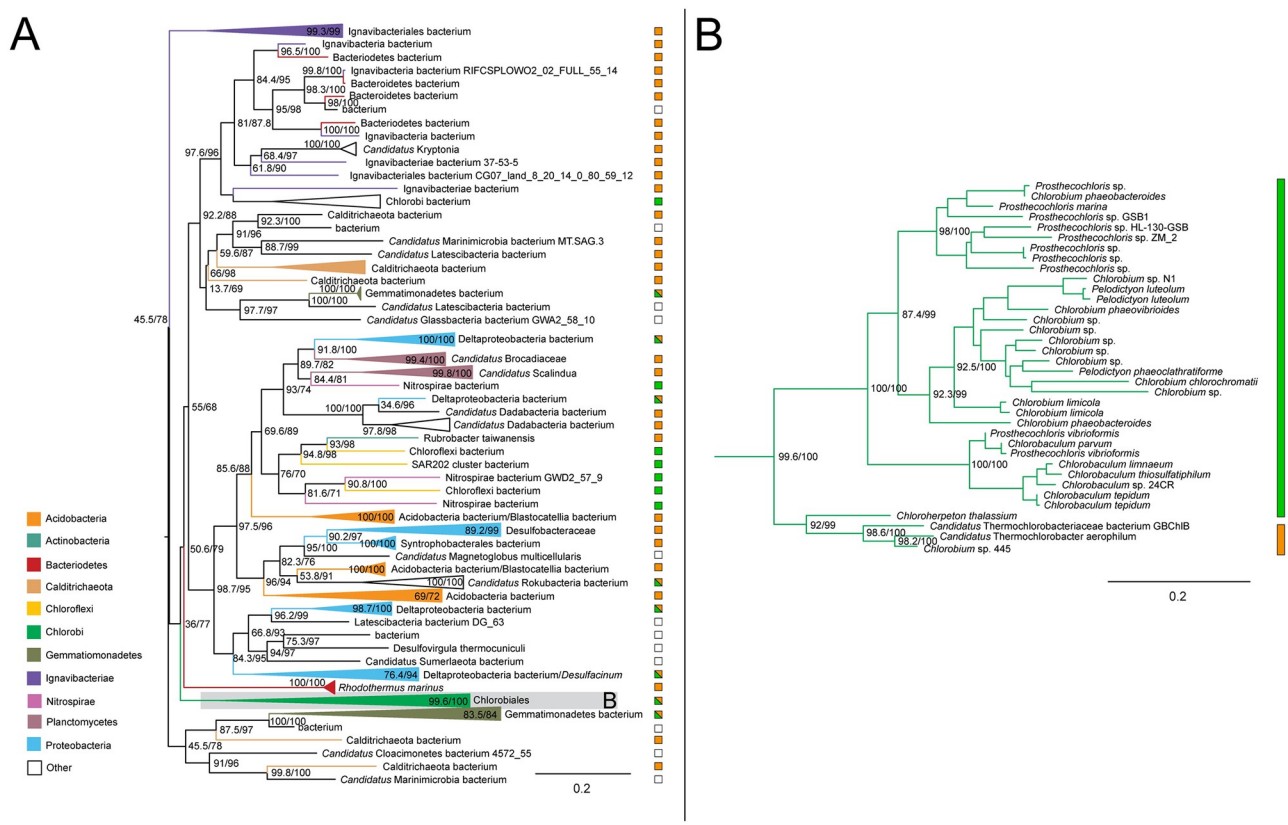

**Fig 3. Maximum likelihood (ML) tree of malate dehydrogenase homologs.** (A) midpoint-rooted tree with collapsed clades labeled with taxonomic group names. (B) Higher resolution tree showing crown Chlorobiales. Support values indicate approximate likelihood ratio test (aLRT)/ bootstrap (100 replicates). Major bipartitions with bootstrap (BS) support are labeled. The full tree contains 242 taxa. Collapsed clades are labeled with taxonomic group names. Color bars or boxes to the right of the trees indicate autotrophic (green), heterotrophic (orange), or undetermined (white) carbon metabolisms. Clades including multiple carbon metabolisms are indicated with diagonally hatched boxes.

**Fumarate hydratase.** Genes encoding fumarate hydratase do not recover the monophyly of Chlorobiales, with the ortholog from *Chloroherpeton thalissium* more closely related to homologs from other bacterial groups. This is consistent with two possible evolutionary scenarios: (1) ancestral presence and vertical inheritance within Chlorobiales (with the exception of *C.thalissium*) in which the gene underwent subsequent orthologous displacement; (2) independent acquisition via two independent HGT events following divergence of *C.thalissium* from the ancestor lineage of other Chlorobiales. In either scenario, the *C.thalissium* copy of the gene was likely acquired from within gammaproteobacteria; some members of Chloroflexi and Acidobacteria appear to have acquired this gene within gammaproteobacteria as well (Fig 4A). *Chlorobium* sp. 445 and *Candidatus Thermochlorobacter*, which usually group with *C.thalissium* on other trees, do not for this gene, which further indicates an orthologous displacement for *C.thalissium* after its divergence from the photoheterotrophic lineages. The broad and sparse taxonomic distribution of sequences grouping with crown Chlorobiales (Fig 4B) prevents the identification of a likely HGT donor group. or an inference of vertical inheritance in stem Chlorobiales, even though some Ignavibacteria sequences are present.

**Fumarate reductase.** The ML tree of fumarate reductase homologs recovers the monophyly of Chlorobiales with the exception of the thermophilic photoheterotrophs *Thermochlorobacter*, and *Chlorobium* sp.445. Observed sequences are closely related to those from

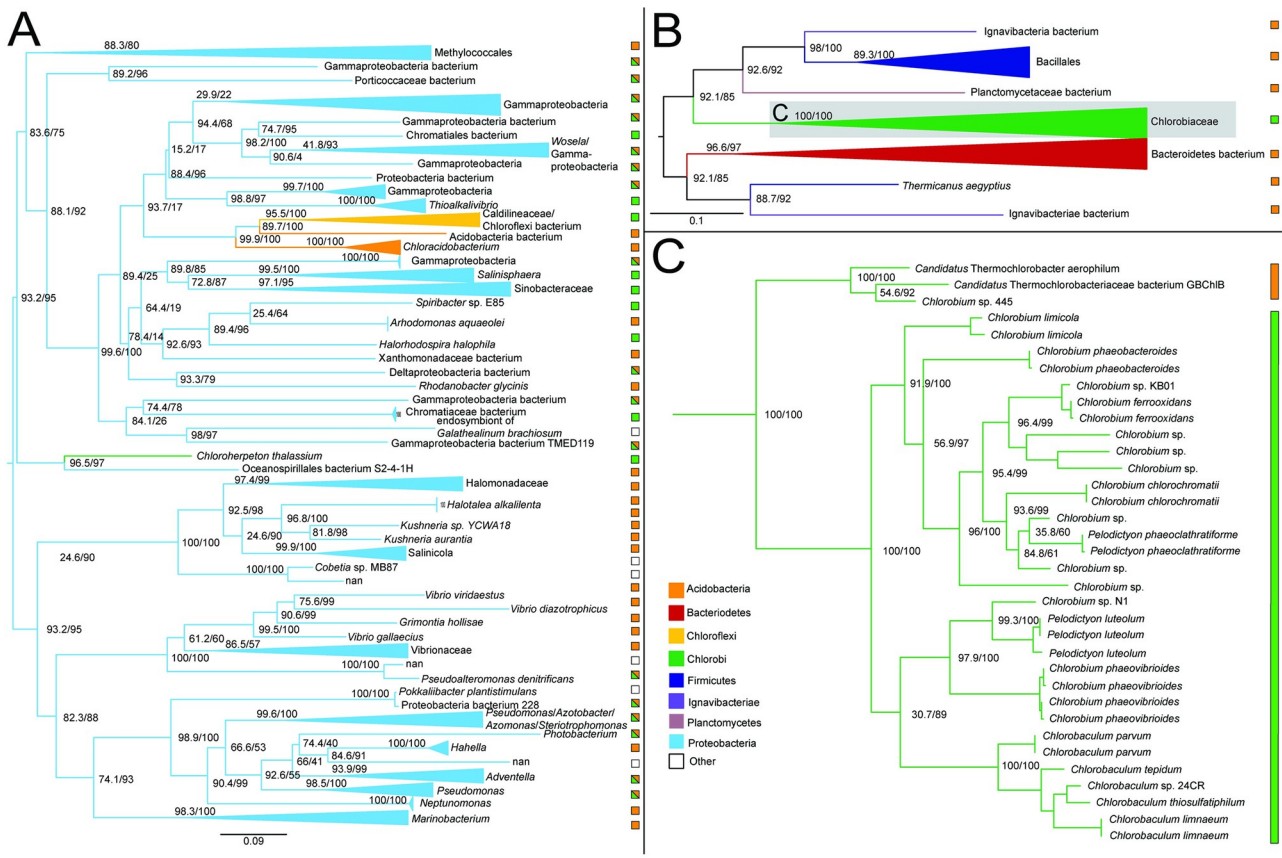

**Fig 4. Maximum likelihood (ML) trees of fumarate hydratase homologs.** (A-B) Two groups of sequences related to Chlorobiales query genes were identified, shown as midpoint rooted trees with collapsed clades labeled with tax¬onomic group names. (C) Higher resolution tree of crown Chlorobiales sequences from B. Support values indicate approximate likelihood ratio test (aLRT)/ bootstrap (100 replicates). Major bipartitions with bootstrap (BS) support are labeled. The full trees contain 250 and 242 taxa, respectively. Color bars to the right of the tree indicate autotrophic (green), heterotrophic (orange), or undetermined (white) carbon metabolisms. Clades including multiple carbon metabolisms are indicated with diagonally hatched boxes.

Candidatus Lambdaproteobacteria, *Geobacter*, and Desulfuromonadales bacterium (Fig 5). These are distantly related by a long branch to other bacterial sequences, including members of Firmicutes, Actinobacteria, and other Proteobacteria, indicative of an HGT into stem Chlorobiales, although the donor group cannot be discerned. The short branch separating the Chlorobiales and proteobacterial sequences indicates a history of relatively recent HGT involving stem Chlorobiales, although the absence of a polarizing outgroup prevents the direction of HGT from being established. A stem Chlorobiales lineage could have been the primary HGT donor to this group of Proteobacteria, or vice versa. Both groups could, alternatively, have been HGT recipients from an unsampled or extinct lineage, potentially explaining the long empty branch preceding this grouping.

**Succinyl-CoA synthetase.** The ML tree for succinyl-CoA synthetase recovers the monophyly of Chlorobiales with a well-supported sibling group including Cytophagales, Marinifilaceae, other Bacteroidetes, and Ignavibacteria (Fig 6). While this outgroup contains similar taxa to the species tree outgroup of Chlorobiales, the phylogeny shows a mixing of Ignavibacteriales and Bacteroidetes groups that prevents inference of vertical inheritance in the Chlorobiales stem ancestor, or identification of an HGT donor group. Photoheterotrophic Chlorobiales

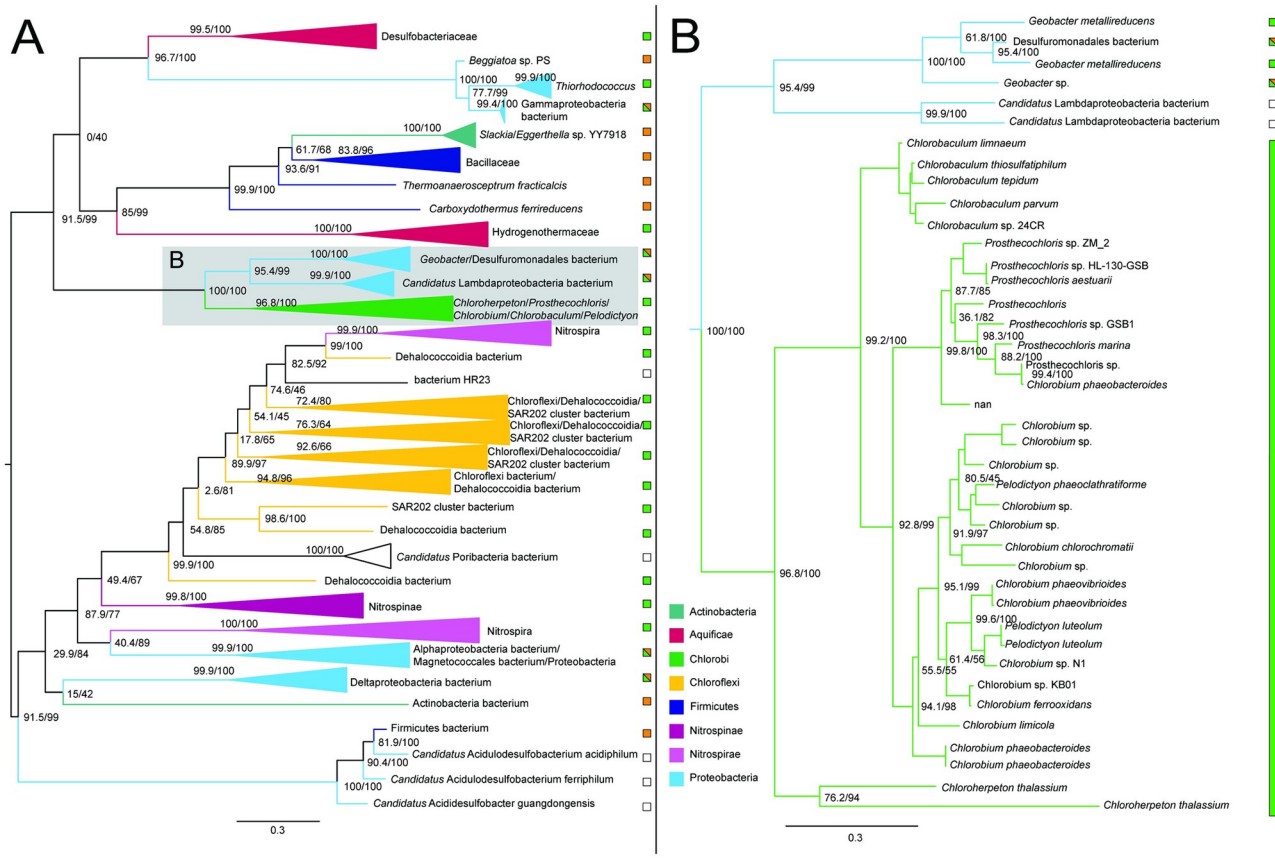

**Fig 5. Maximum likelihood (ML) tree of fumarate reductase homologs.** (A) midpoint-rooted tree with collapsed clades labeled with taxonomic group names. (B) Higher resolution tree of crown Chlorobiales and closely related proteobacterial sequences. Support values indicate approximate likelihood ratio¬¬ test (aLRT)/ bootstrap (100 replicates). Major bipartitions with bootstrap (BS) support are labeled. The full tree contains 248 taxa. Color bars to the right of the tree indicate autotrophic (green), heterotrophic (orange), or undetermined (white) carbon metabolisms. Clades including multiple carbon metabolisms are indicated with diagonally hatched boxes.

have also retained this gene, indicating that its presence is not strictly indicative of a functional rTCA cycle. Succinyl-CoA may be involved in any number of additional metabolic functions, including heme synthesis, ketone metabolism, and full or partial oxidative TCA cycles [39]. In bacteriochlorophyll biosynthesis, succinate/succinyl-CoA can be carboxylated by a partial rTCA cyle to create 2-oxoglutate as precursor molecules, providing a pre-adaptation in the form of a selective advantage to photoheterotrophic lineages [35, 40]. More distantly related outgroups on the tree also show extensive HGT between bacterial groups, including Calditrichaeota, Firmicutes, Acidobacteria, and Proteobacteria.

**2-oxoglutarate: Ferredoxin oxidoreductase.** The tree for 2-oxoglutarate: ferredoxin oxidoreductase alpha subunit recovers the monophyly of Chlorobiales, grouping within a taxonomically broad range of Bacteroidetes (Fig 7). The placement of the specific orders Chitinophaga, Flavobacteriales, Sphingobacteriales, and Balneolaceaes within the tree suggests similarity to published species trees of Bacteroidetes [33], and is further indicative of a likely HGT from within Bacteroidetes. The ML tree does not recover the sibling grouping of *C.thalassium* and thermophilic photoheterotrophic lineages, suggesting a possible misrooting of Chlorobiales in this tree.

**Isocitrate dehydrogenase.** The isocitrate dehydrogenase phylogeny recovers a polyphyletic placement of sequences within Chlorobiales, with *Chlorobium*, *Pelodictyon*, and

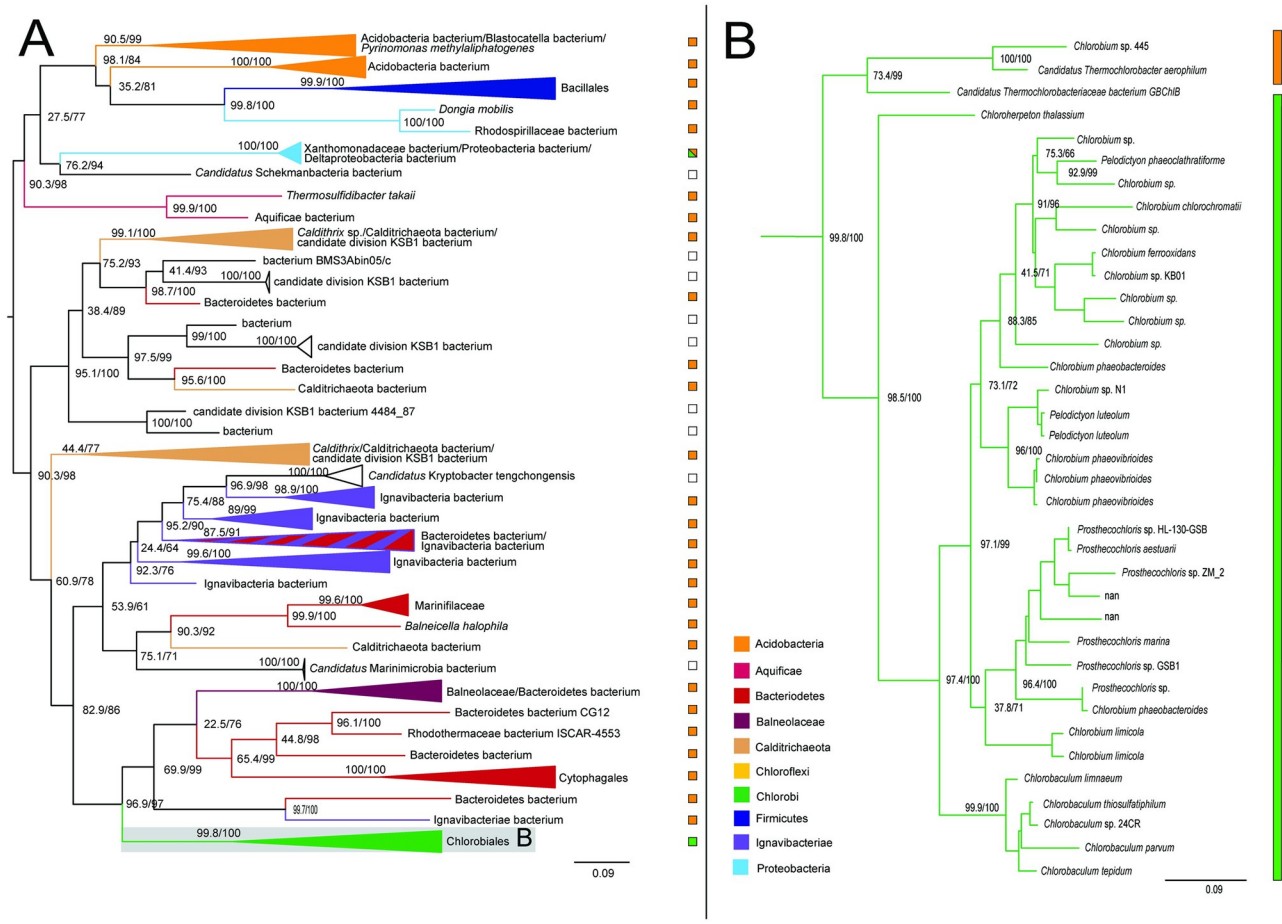

**Fig 6. Maximum likelihood tree of succinyl-CoA synthetase homologs.** (A) midpoint-rooted tree with collapsed clades labeled with taxonomic group names. (B) Higher resolution tree of crown Chlorobiales. Support values indicate approximate likelihood ratio test (aLRT)/ bootstrap (100 replicates). Major bipartitions with bootstrap (BS) support are labeled. The full tree contains 248 taxa. Color bars to the right of the tree indicate autotrophic (green), heterotrophic (orange), or undetermined (white) carbon metabolisms. Clades including multiple carbon metabolisms are indicated with diagonally hatched boxes.

*Chlorobaculum* genera and *Prosthecochloris* genera forming two distinct groups within one set of identified homologs (Fig 8A) and *C.thalassium*, *Thermochlorobacter*, and *Chlorobium* sp.445 group within another set of identified homologs (Fig 8B), providing clear evidence of multiple HGT events for this gene family. Specific HGT donor groups to Chlorobiales cannot be clearly identified from these trees. The placement of closely related Chlorobiales genera in Fig 8A is consistent with vertical inheritance in this group, with subsequent multiple HGTs to other bacterial groups, including Deltaproteobacteria, Desulfobulbaceae bacterium, and Marinifilaceae. These distinct gene histories suggest that the clades within Chlorobiales shown in Fig 8C and Fig 8D acquired isocitrate lyase independently after they diverged.

**Aconitate hydratase.** Aconitate hydratase sequences within Chlorobiales are polyphyletic with *C. thalassium, Thermochlorobacter*, and remaining taxa each grouping with different sets of bacterial sequences. The main group (Fig 9C) places within Epsilonproteobacteria, suggesting this is the HGT donor group to the major clade of crown Chlorobiales as well as the HGT donor to a subset of Aquificales, a frequency observed HGT partner with Epsilonproteobacteria. However, long branches and sparse taxonomic representation prevent a reliable rooting

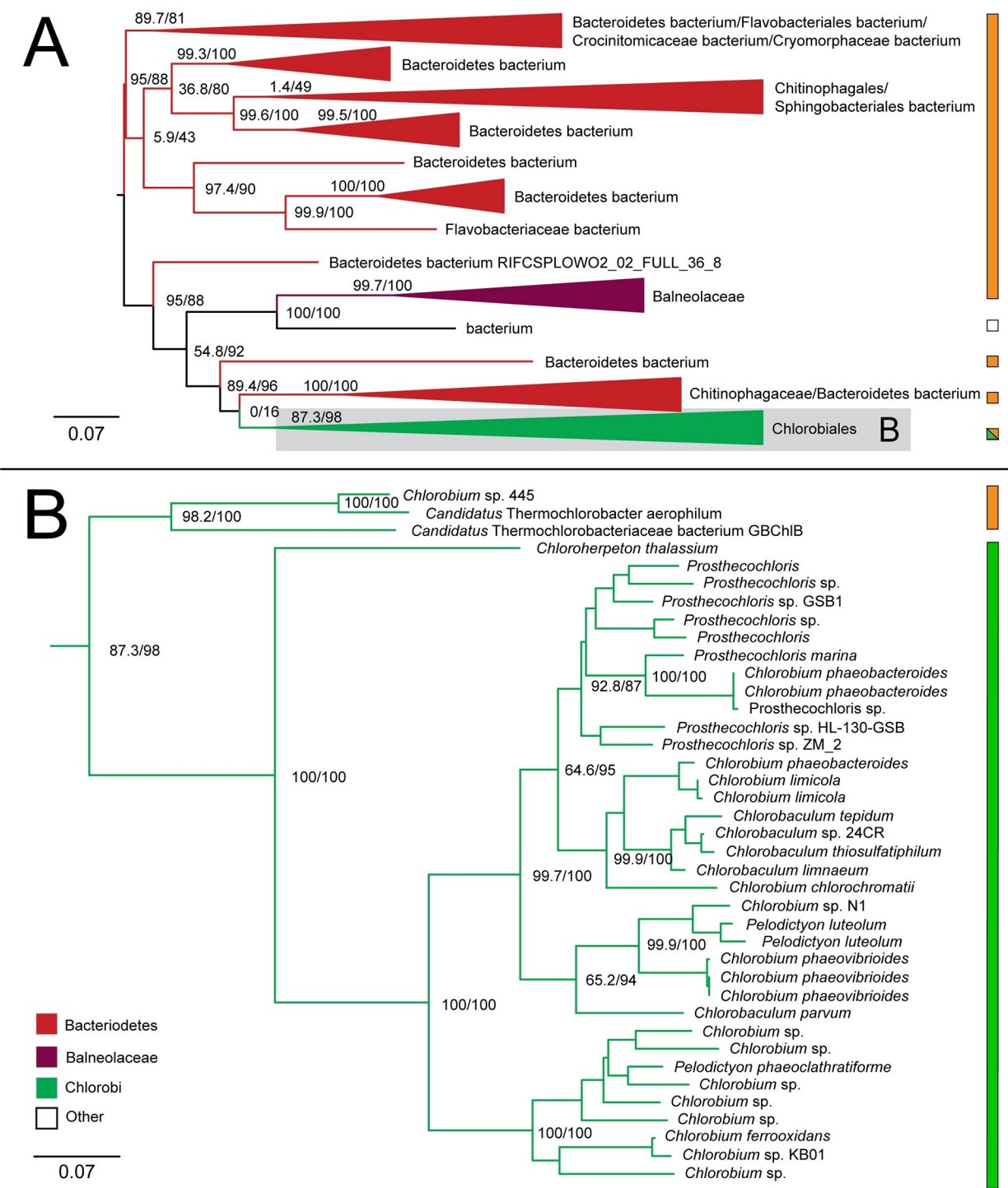

**Fig 7. Maximum likelihood (ML) tree of 2-oxoglutarate: Ferredoxin oxidoreductase homologs.** (A) midpoint-rooted tree with collapsed clades labeled with taxonomic group names. (B) Higher resolution tree of crown Chlorobiales. Support values indicate approximate likelihood ratio test (aLRT)/ bootstrap (100 replicates). Major branches with bootstrap (BS) support are labeled with respective values. Tree A contains 243 taxa. Color bars to the right of the tree indicate autotrophic (green), heterotrophic (orange), or undetermined (white) carbon metabolisms. Clades including multiple carbon metabolisms are indicated with diagonally hatched boxes.

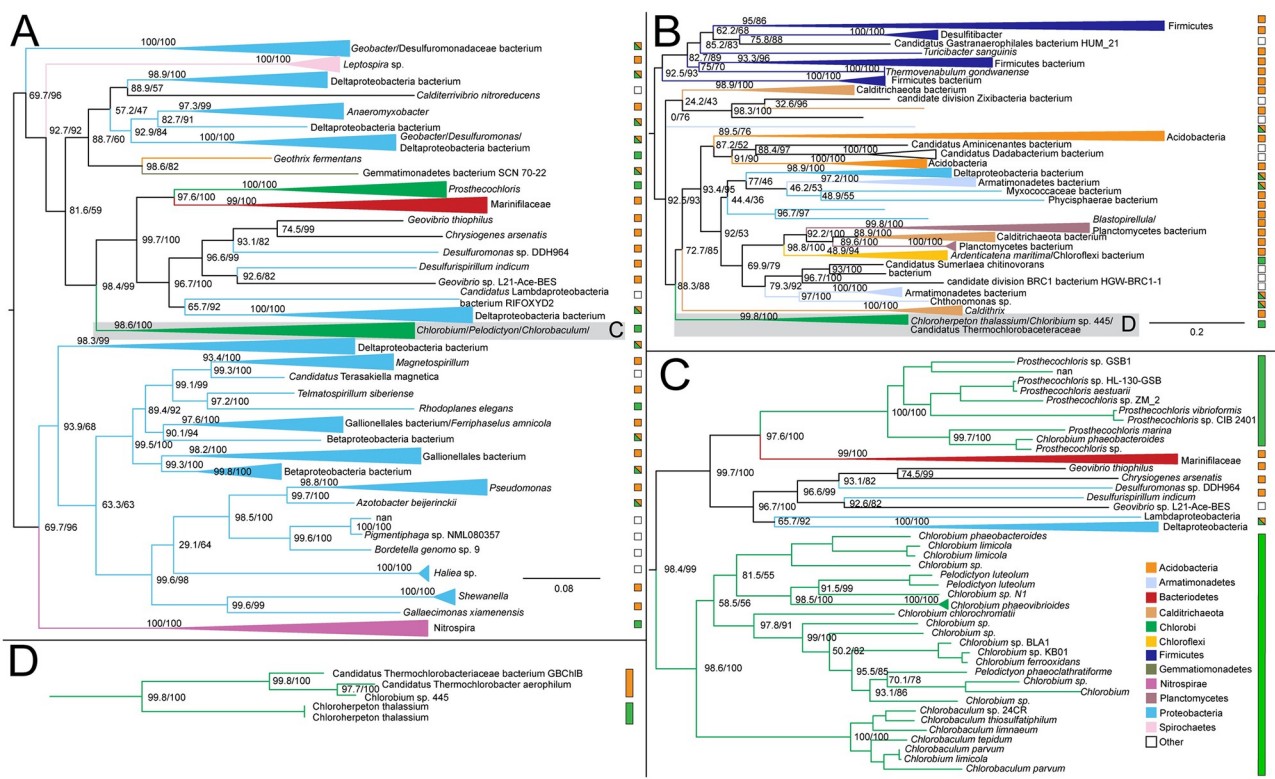

**Fig 8. Maximum likelihood (ML) trees of isocitrate dehydrogenase homologs.** Two groups of sequences related to Chlorobiales query sequences were identified, shown as midpoint rooted trees with collapsed clades labeled with taxonomic group names (A-B). (C-D) Higher resolution trees of crown Chlorobiales groups. Support values indicate approximate likelihood ratio test (aLRT)/ bootstrap (100 replicates). Major bipartitions with bootstrap (BS) support are labeled. A and B trees contain 250 and 232 taxa, respectively. Color bars to the right of the tree indicate autotrophic (green), heterotrophic (orange), or undetermined (white) carbon metabolisms. Clades including multiple carbon metabolisms are indicated with diagonally hatched boxes.

and clear identification of a donor clade within Epsilonproteobacteria. The absence of aconitate hydratase within other photoheterotrophic lineages can be explained by two possible scenarios: (1) initial acquisition in the common ancestor lineage of Chloroherpetonaceae, with subsequent loss in the photoheterotroph ancestor; or (2) acquisition by *C. thalassium* after the divergence of the photoheterotroph ancestor lineage. Both scenarios include a subsequent independent acquisition by *Thermochlorobacter*.

In the case of (1) the photoheterotroph clade can be inferred to be ancestrally autotrophic, as all rTCA cycle genes would be present in their common ancestor; in the case of (2) their common ancestor would be inferred to be lacking a complete rTCA cycle, and therefore extant photoheterotrophic lineages would represent a continuation of an ancestral metabolic state. However, this latter scenario is less likely, as all rTCA enzymes but aconitate hydratase would be ancestrally present, a "nearly complete" rather than merely a "partial" rTCA cycle that would not have a known modern analog.

**ATP citrate lyase.** More than other enzymes, ATP citrate lyase is considered diagnostic for the presence of rTCA cycle carbon fixation, as it plays the crucial step of citrate cleavage into acetyl-CoA and oxaloacetate [41]. A variant of this process is found in Aquificae, where cleavage is catalyzed in tandem by citryl-CoA synthetase and citryl-CoA lyase [42]. These enzymes share a distant common ancestry indicated by sequence homology as well as

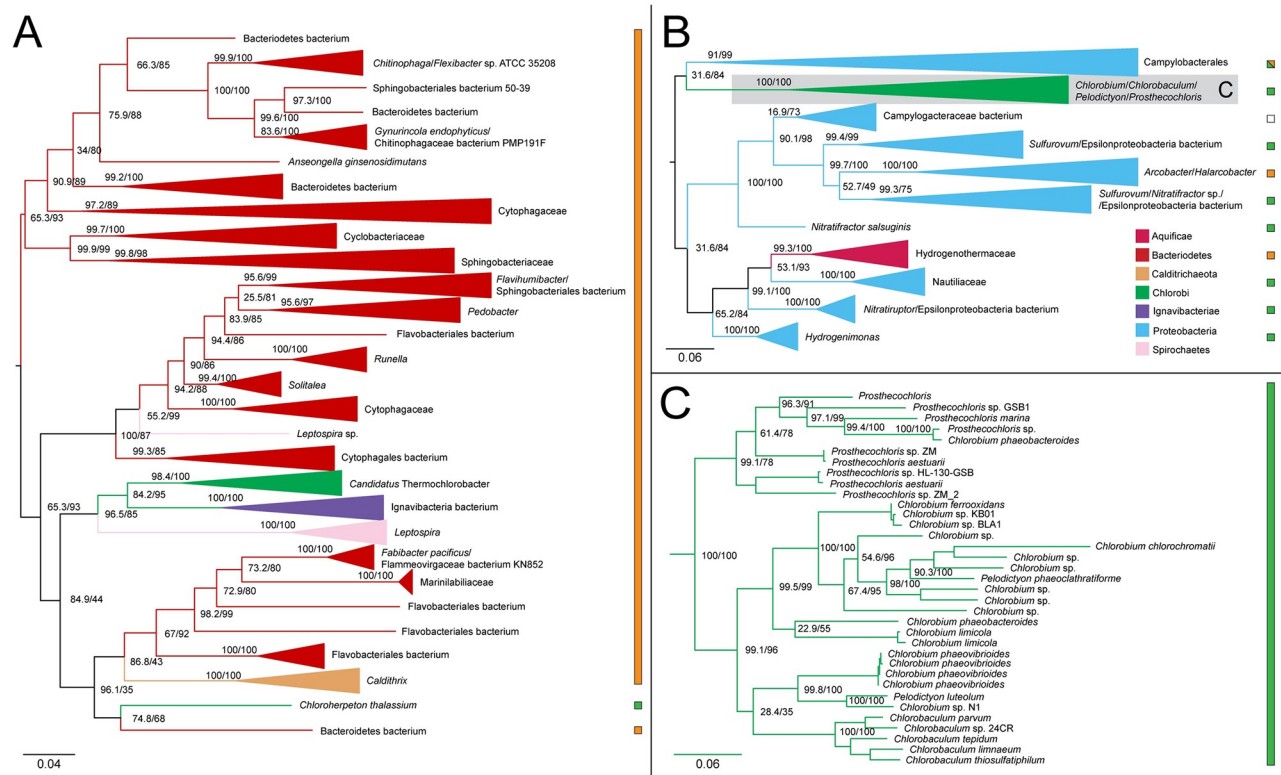

**Fig 9. Maximum likelihood (ML) tree of aconitate hydratase homologs.** Two groups of sequences related to Chlorobiales query sequences were identified, shown as midpoint rooted trees with collapsed clades labeled with taxonomic group names (A-B). A higher resolution tree for Chlorobiales sequences in tree B is shown in (C). Support values indicate approximate likelihood ratio test (aLRT)/bootstrap (100 replicates). Major bipartitions with bootstrap (BS) support are labeled. A and B trees contain 249 and 250 taxa, respectively. Color bars to the right of the tree indicate autotrophic (green), heterotrophic (orange), or undetermined (white) carbon metabolisms. Clades including multiple carbon metabolisms are indicated with diagonally hatched boxes.

conserved structural elements to the two subunits of ATP citrate lyase in *Chlorobium limicola*, specifically, a two-helix stalk and β-hairpins [43] (S2 Fig).

In our results, all autotrophic Chlorobiales form a single group in the ML phylogeny of both the α (Fig 10) and β (S1 Fig) subunits of this protein, which includes Nitrospirae and Nitrospinae sequences, sibling to the deeply branching heterotrophic *Chlorobium* sp. 445 lineages (Fig 10B). The direction of HGT between Chlorobiales and Nitrospirae/Nitrospinae is unclear, as the recovered root of Chlorobiales is inconsistent with the species tree and varies between subunits. However, the α subunit sequence within *Chlorobium* sp. 445 is only partial, missing 167 amino acids (S2 Fig). This absence in the alignment may impose tree reconstruction artifacts, resulting in the spurious rooting of this group, complicating HGT inference. To test this, we re-ran the ML tree using only the sequence region present in *Chlorobium* sp. 445. This recovered the same phylogenetic placement of sibling groups Nitrospirae/Nitrospinae with respect to Chlorobiales. Furthermore, Bayesian inference, an approach more resistant to long branch attraction artifacts, also recovered the same rooting for both subunits (S3 Fig). If not mis-rooted, this tree topology could also be explained by an ancestral loss of ATP citrate lyase in the photoheterotroph ancestor, followed by a secondary acquisition in *Chlorobium* sp. 445, either from the Nitrospirae/Nitrospinae lineage, or an HGT donor group common to both, although this is a less parsimonious scenario.

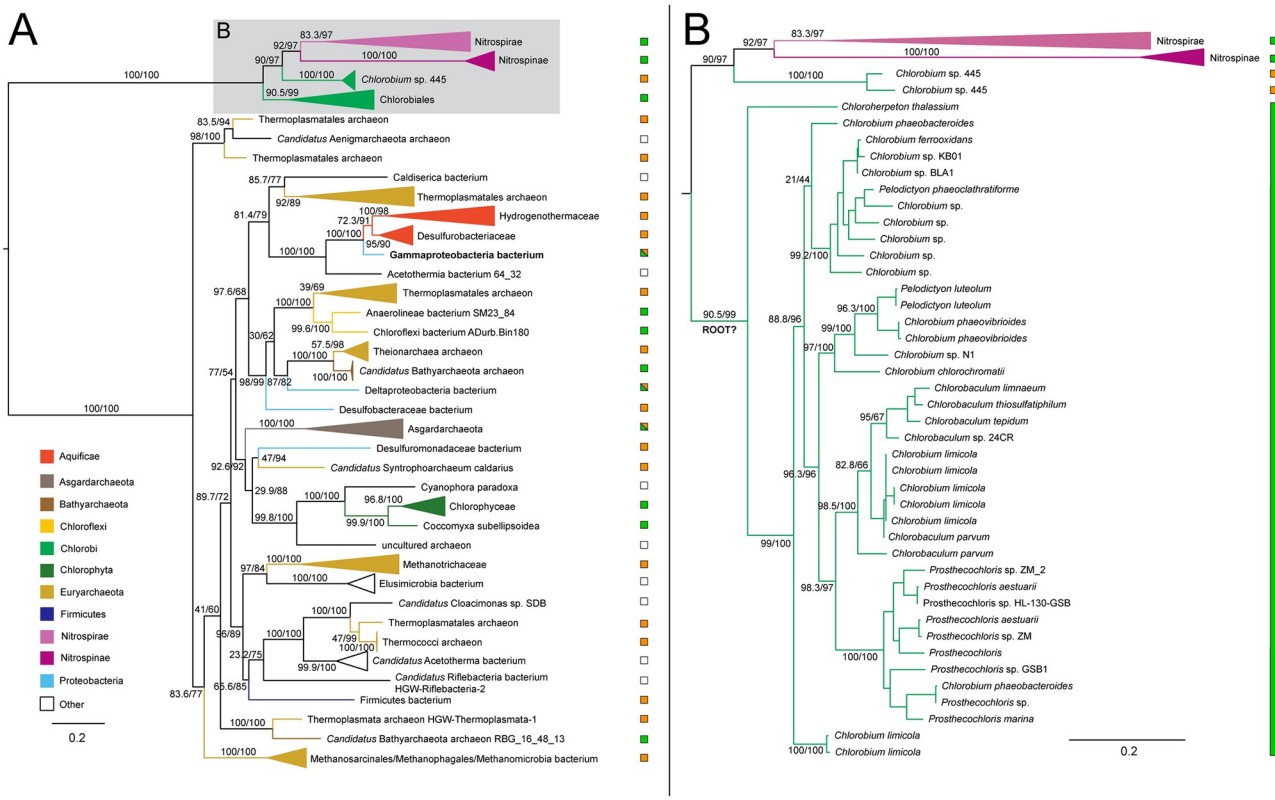

**Fig 10. Maximum likelihood (ML) tree of ATP citrate lyase alpha subunit homologs.** (A) Midpoint-rooted tree with collapsed clades labeled with taxonomic group names. (B) Higher resolution tree showing crown Chlorobiales and closely related sequences in Nitrospirae/Nitrospinae. Support values indicate approximate likelihood ratio test (aLRT)/ bootstrap (100 replicates). Major clades with bootstrap (BS) support are labeled with respective values. The full tree contains 247 taxa. Color bars to the right of the tree indicate autotrophic (green), heterotrophic (orange), or undetermined (white) carbon metabolisms. Clades including multiple carbon metabolisms are indicated with diagonally hatched boxes.

A mapping of this missing region to the crystal structure of the ATP citrate lyase enzyme structure shows a missing a stabilizing β hairpin region (S2 Fig), suggesting that this partial version may be nonfunctional, or retain a function unrelated to carbon fixation. This may also be evidence of independent loss of rTCA carbon fixation capability within other photohetero-trophic lineages, such as *Thermochlorobacter*. Still, without knowing if the truncated variant is functional, it brings the viability of ATP citrate lyase as a marker for autotrophy into question.

Other published phylogenies of ATP citrate lyase shows a history consistent with our find-ings, with vertical inheritance in Chlorobiales, but concluding that Nitrospirae and Nitrospi-nae horizontally transferred these genes to Chlorobiales, albeit with low statistical support [44]. The long branch separating these sequences from other taxa in the tree obscures the ancestry of ATP citrate lyase within the Chlorobiales stem lineage, and no clear HGT donor group can be inferred. The absence of other closely related sequences suggests an acquisition from unsampled or extinct groups. Further analyses are thus required to resolve the true his-tory of these genes.

**Pyruvate: Ferredoxin oxidoreductase.** The ML tree for pyruvate: ferredoxin oxidoreduc-tase shows very sparse taxonomic outgroups separated by a long branch indicative of indepen-dent HGT events. Distantly related groups include Bacteroidetes, Ignavibacteria, autotrophic Cyanobacteria, Chloroflexi, Verrucomicrobia, and Acidobacteria (Fig 11A). This enzyme is

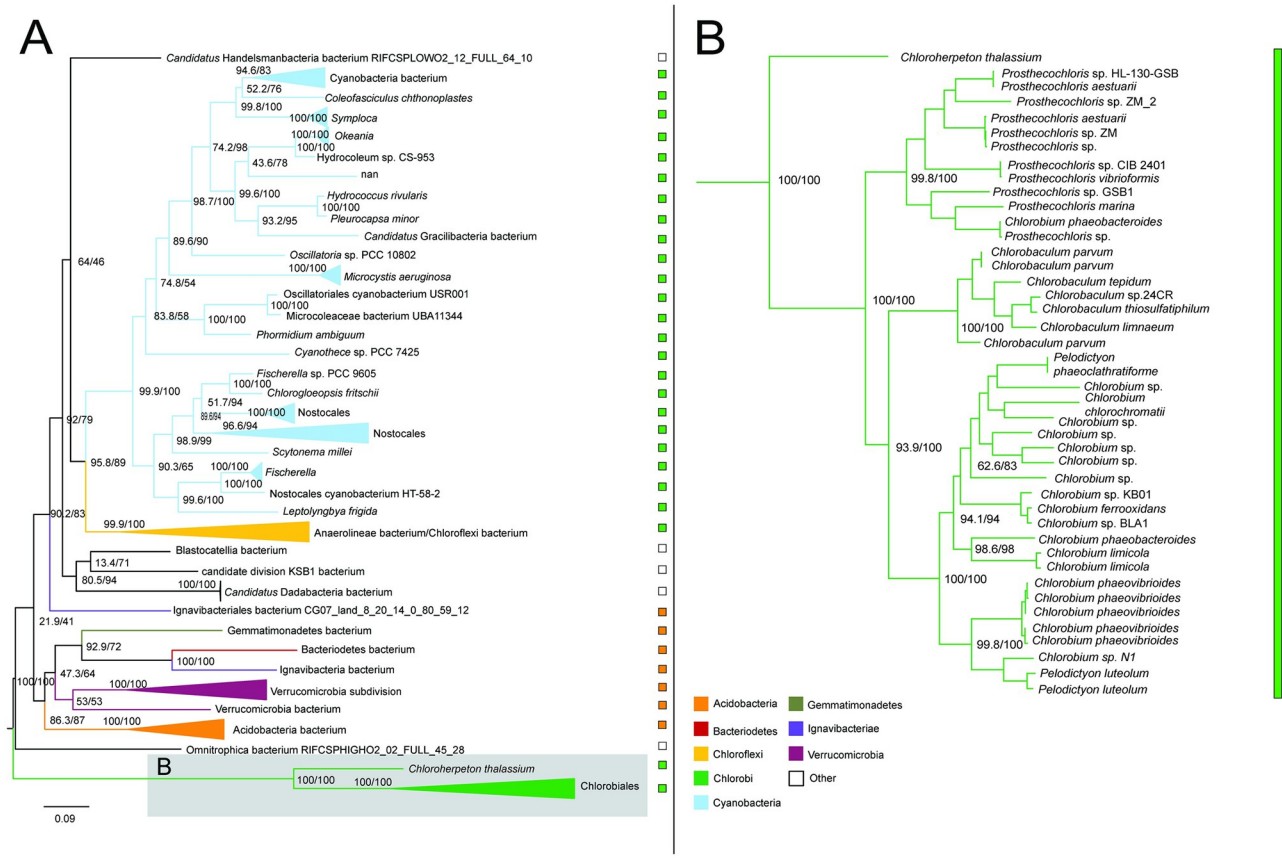

**Fig 11. Maximum likelihood (ML) tree of pyruvate: Ferredoxin oxidoreductase homologs.** (A) Midpoint-rooted tree with collapsed clades labeled with taxonomic group names. (B) Higher resolution tree showing crown Chlorobiales and other closely related sequences. Support values indicate approximate likelihood ratio test (aLRT)/ bootstrap (100 replicates). Major clades with bootstrap (BS) support are labeled with respective values. The full tree contains 247 taxa. Color bars to the right of the tree indicate autotrophic (green), heterotrophic (orange), or undetermined (white) carbon metabolisms. Clades including multiple carbon metabolisms are indicated with diagonally hatched boxes.

one of only three that are present in the rTCA cycle and not the oxidative TCA cycle [24] and was apparently lost within the photoheterotrophic Chlorobiaceae lineages, as they do not appear with other Chlorobiales in the gene tree (Fig 11B). Due to branch lengths, a putative donor group cannot be established.

## rTCA cycle proteins do not trace the evolutionary history of autotrophy

Additional evidence for the patchwork assembly of the rTCA cycle within Chlorobiales is provided by the taxonomic distribution of autotrophy across species represented in each protein tree. Surprisingly, in the majority of cases the most closely related outgroup sequences are not from autotrophic species, suggesting that these proteins have an alternative function in these groups (fumarate hydratase, succinyl-CoA synthetase, OGOR). Presumably, before the complete rTCA pathway was present, these proteins performed similar functions in Chlorobiales as part of a photoheterotrophic metabolism.

Further evidence of the potential utility of incomplete sets of rTCA cycle proteins is provided by the retention of several of these proteins within photoheterotrophic Chlorobiales; in some cases, it appears these proteins were acquired after the divergence from other Chlorobiaceae (fumarate hydratase, isocitrate lysase, aconitate hydratase) while others were vertically

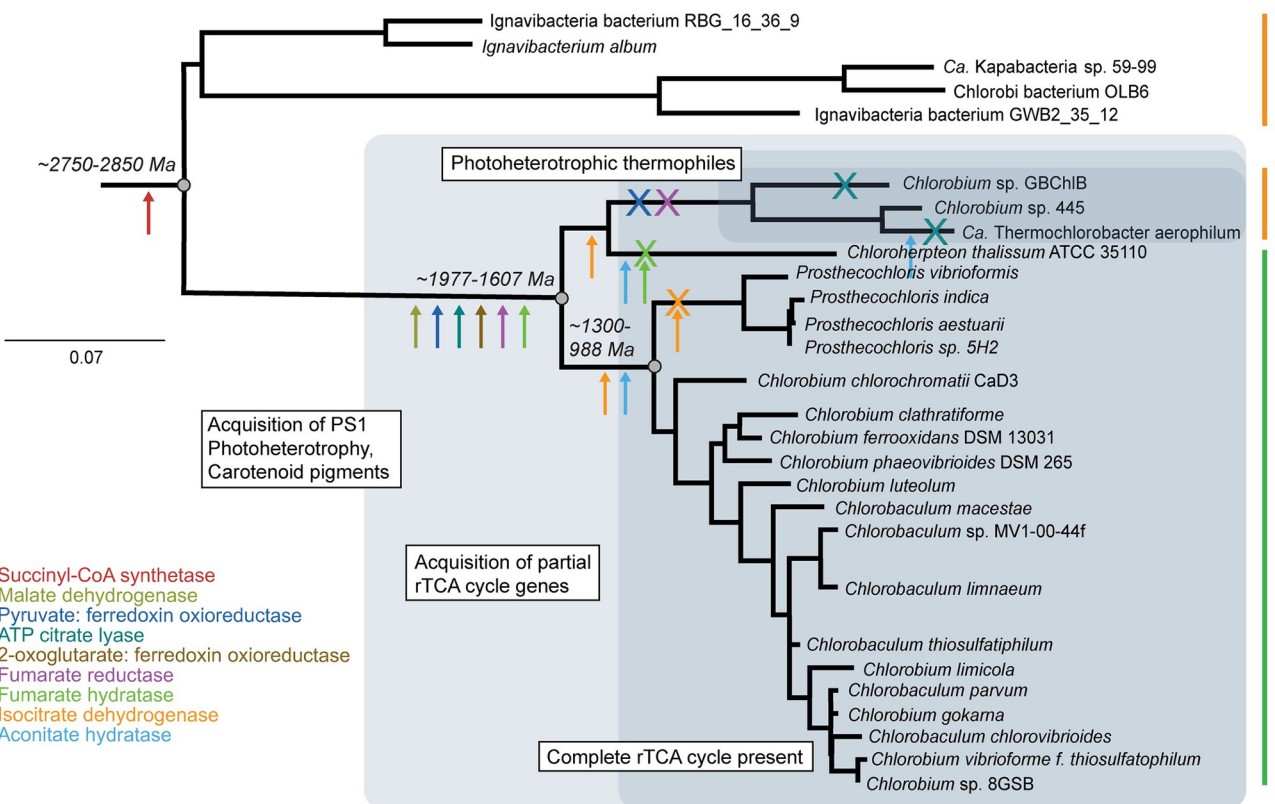

**Fig 12. 16S rRNA tree of Chlorobiales and Ignavibacterium (outgroup) with gene histories of rTCA proteins mapped.** Arrows indicate inferred acquisitions of specific enzymes, and subsequent losses are indicated with crosses with enzyme names coded by color. Shaded fields indicate inferred carbon metabolisms for different clades. Divergence times labelled by respective nodes [13].

inherited within the crown group (malate dehydrogenase, succinyl-CoA synthetase, 2-oxoglutarate: ferredoxin oxidoreductase). In all these cases, the genes present are also utilized in the oxidative TCA cycle, thus potentially explaining their retention in the divergent thermophiles. Elucidating their functions within these groups may provide clues to the ancestral functions of these genes in stem Chlorobiales, before a complete rTCA cycle was present. The inferred presence of ATP citrate lyase in stem Chlorobiales, in the absence of a complete and functional rTCA cycle, further suggests that this enzyme is not, on its own, diagnostic for autotrophy in this group, as is observed for other heterotrophic microbes [45, 46].

## Discussion

### A Chimeric origin of rTCA cycle genes and autotrophy within Chlorobiales

Our results show a striking diversity of the evolutionary histories of rTCA cycle enzymes within Chlorobiales, suggesting that carbon fixation has a complex and relatively recent evolutionary history within this group (Fig 12). There is strong evidence for the independent acquisition of several enzymes via HGT from different donors, both to the stem group, and to different lineages within Chlorobiales following their crown divergence. While current taxon sampling and the inherent limitations of phylogenetic inference prevent, in most cases, the identification of specific HGT donor groups for these genes, their collective histories are unambiguously incompatible with the alternative hypothesis of vertical inheritance from stem

Chlorobiales. This interpretation is consistent with previous phylogenetic trees in which the inheritance of rTCA enzymes could not be narrowed down to a specific donor group or transfer event [44, 47].

In several cases, *C. thalassium* and thermophilic organisms have orthologs of rTCA genes not closely related to those in other members of Chlorobiales, clearly indicating a more recent history of HGT for one or more subgroups. From these observations, two potential hypotheses emerge regarding the evolution of the rTCA cycle in Chlorobiales: (1) the full rTCA cycle was acquired in the stem lineage, with some enzymes replaced via additional HGTs in descendant crown group lineages; or (2) the full rTCA cycle was only assembled later, within crown Chlorobiales, with independent acquisition of some enzymes by multiple lineages. We find the latter explanation to be more parsimonious. Only for the fumarate hydratase tree is (1) a reasonable hypothesis (Fig 4A). When *Candidatus Thermochlorobacter* and *Chlorobium* sp. 445 appear with *C. thalassium* in the isocitrate lysase (Fig 8A) and aconitate hydrase (Fig 9A) trees, this implies (2) an independent acquisition via two HGT events following the divergence within the family of Chlorobi. These trees each have variable outgroups which include both autotrophic and heterotrophic bacteria/archaea.

## rTCA cycle genes within photoheterotrophic lineages

A functional rTCA cycle was likely originally present in the common ancestor of photoheterotrophic thermophilic Chlorobiales lineages. In several gene trees showing HGT within crown Chlorobiales, these group together with *C. thalassium*, indicating a likely shared ancestral state of autotrophy. Other genes present in a complete rTCA path were apparently lost in thermophilic members, consistent with their photoheterotrophic physiology. For example, *Candidatus Thermochlorobacter aerophilum* and *Chlorobium sp.* GBChlB do not have genes encoding ATP citrate lyase, an essential enzyme in the pathway (Fig 10B). Alternative genes to ATP citrate lyase that may rescue rTCA cycle function (citryl-CoA synthetase and citryl-CoA lyase) [42] are also absent. The retention of a partial ATP citrate lyase within *Chlorobium* sp. 455 further suggests that these genes were independently lost within thermophilic lineages following their diversification.

Other genes of the rTCA cycle were apparently retained in this group, suggesting functional roles independent of carbon fixation. It is also possible that 2-oxoglutarate:ferredoxin and pyruvate:ferredoxin might be involved in mixotrophic growth in an incomplete rTCA path in these lineages [34, 35, 48].

## Molecular clocks inform the history of carbon fixation within Chlorobiales

The existence of the rTCA cycle as a primordial carbon fixation pathway, mediated by nonenzymatic, inorganic catalysts, is a hotly debated issue in origins of life research [20, 49, 50]. It has also been proposed that an ancestral rTCA was present in an autotrophic last universal common ancestor (LUCA) [51, 52]. However, the phylogenomic signal needed to establish this deep ancestry may be impossible to fully elucidate, given the relatively shallow taxonomic distributions of rTCA cycle constituent proteins, and high frequency of HGT between phyla, as evidenced by reconstructed gene trees. Nevertheless, the clear history of rTCA cycle proteins within Chlorobiales, and the fidelity of their inheritance in several subgroups, permits the potential dating of carbon fixation within this clade. Recent molecular clock studies have produced age estimates for major divergences within Chlorobiales informed by large protein sequence datasets, cyanobacterial fossil calibrations, and HGT constraints [13]. These ages show crown Chlorobiales diversifying between 1977 Ma and 1607 Ma, and subsequent diversification of Chlorobiaceae between 1300 Ma and 988 Ma (Fig 12. While these age uncertainties

**Table 2. Isotopic differences by carbon fixation pathways and taxa [54–64].**

| Pathway | Taxa | $\Delta\delta 13C\%$ |
|---|---|---|
| CBB | Cyanobacteria, Eukaryota, Proteobacteria (Alpha,Beta,Gamma), Oschillochloridaceae | $-10 \rightarrow -35$ |
| rTCA | Chlorobiales, Aquificales, Nitrospirae, Nitrospinae | $-2 \rightarrow -23$ |
| Wood-Ljungdahl | Euryarcheota (methanogens), Planctomycetes, Deltaproteobacteria, Spirochaetes | $-5 \rightarrow -80$ |

Table notes: Sourced as $\Delta\delta 13C = \delta 13C_{biomass} - \delta 13C_{carbon\ source}$ calculation. Listed taxa are not diagnostic of full pathway distribution, but major groups of relevance. There is a 25% mean deviation between inorganic and organic carbon isotopes [65].

are large and will likely be revised in future studies with additional constraints, they are consistent with BCF lipid biomarkers at 1.64 Ga most likely being produced after the crown diversification of Chlorobiales but before the diversification of major extant autotrophic groups. These phylogenetic analyses therefore raise the intriguing possibility that BCF lipid biomarkers were produced by early diverging lineages that did not yet have a complete rTCA cycle. If that is indeed the case, they would not preserve C isotopic fractionations consistent with Chlorobiales carbon fixation under rTCA ($\Delta\delta 13C$ -20 to -10%) [53], but rather should show fractionations more like those observed for extant photoheterotrophic lineages, and/or BCF bulk organic carbon (Table 2). Furthermore, an isotopic shift in preserved biomarkers to modern values reflective of carbon fixation via an rTCA cycle should be expected to occur before 988 Ma [13]. These interpretations may be complicated by the existence of unsampled or extinct Chlorobiales stem groups that preserved an ancestral heterotrophic metabolism long after the emergence of autotrophy within more derived groups; however, it is unlikely that such groups would be ecologically abundant following selection favoring autotrophy in these environments. Furthermore, aromatic carotenoid synthesis is absent within *Chloroherpeton* and photoheterotroph lineages, suggesting that the most parsimonious history of these genes would have them acquired after the crown divergence of Chlorobiaceae. This argues against the BCF biomarkers being sourced from stem group Chlorobiales, unless there were subsequent losses of biomarker synthesis genes within the crown group.

## Autotrophy in Chlorobiales: Adaptation to a changing Mesoproterozoic world?

The independent assembly of a complete rTCA cycle for carbon fixation in at least two major lineages of Chlorobiales is potentially indicative of broad changes in environmental conditions and ecological niches during the Mesoproterozoic. Because of their unique metabolisms and chemical requirements, Chlorobiales are useful indicators of past environmental conditions [66], most notably for the changes in sulfur abundance and redox state through Earth history. The Mesoproterozoic was a time of complex interactions between sulfur and oxygen cycles, potentially spurring the metabolic evolution of anoxygenic phototrophs such as Chromaticeae and Chlorobiales. Redox-sensitive element abundances and stable isotopes indicate progressive oxygenation of the ocean–atmosphere system, along with a significant increase in marine sulfate inventory during this time [1, 2, 67–69]. This increased oxygenation of the atmosphere promoted weathering of pyrite and delivery of sulfate to aquatic environments, subsequently increasing the prevalence of euxinic conditions via microbial sulfate reduction [70, 71]. At the same time, increased oxygenation of surface waters would drive anoxygenic phototrophs

deeper in water columns where these anoxic conditions could persist; these lower light environments would select for more efficient light-harvesting systems, such as chlorosomes [72], and more energy efficient metabolic pathways for processing carbon. This may also explain why Chlorobiales and Chromatiales don't have shared carbon fixation pathways despite their similar physiology and environmental occurrence. Chromatiales are more oxygen-tolerant and can exist higher in the water column where light is more available for energy-intensive carbon fixation via the CBB cycle, requiring seven ATP per molecule of synthesized pyruvate, compared with only two for the rTCA cycle [41].

In such a scenario, the HGT donors of rTCA cycle genes to Chlorobiales would also have inhabited these deeper aquatic environments. Unfortunately, the phylogenies of rTCA cycle proteins do not identify putative HGT donor groups with sufficient resolution to evaluate this hypothesis with the present dataset. Future environmental sequencing investigations of stratified marine environments may discover missing microbial diversity that better resolves these gene tree histories; alternatively, these microbial groups and their metabolisms may have been unique to stratified Proterozoic marine conditions, and have since gone extinct, or have lost these genes. While this scenario may explain why Chlorobiales acquired carbon fixation via rTCA cycle genes rather than CBB cycle genes, it does not, on its own, explain why the shift to autotrophy occurred during this time. One possibility may once again relate to increasing oxygen levels. If aquatic aerobic/sulfate reducing metabolisms also experienced a diversification and ecological expansion at this time, then, presumably, more organic carbon would be microbially oxidized, leaving less available for assimilatory heterotrophic processes [73]. These conditions would therefore favor more widespread autotrophy, even among energy-limited organisms such as Chlorobiales.

## Conclusion

Our proposed scenario for the history of autotrophy within Chlorobiales infers early members of this group were heterotrophs and that autotrophy was independently acquired within crown Chlorobiales lineages through a chimeric history of acquisition of rTCA genes. Therefore, early members of Chlorobiales fractionated carbon differently than extant autotrophic members that have a complete rTCA cycle, a transition that is predicted to be apparent within the isotopic record of lipid biomarkers. This specifically implies that a "GSB-like" carbon isotopic fractionation within preserved aromatic carotenoid biomarkers should not be the standard of evidence to infer Chlorobiales as their biological source. Rather, we would expect that older carotenoid material, such as that obtained from the BCF, should instead show fractionations consistent with heterotrophy, closer to that of bulk organic carbon within the system. This is especially important given the proposed alternative cyanobacterial origin for these lipids, which may be falsely inferred in the absence of observing the expected "rTCA" signature fractionation. Molecular clock studies of Chlorobiales constrain the timing of these metabolic evolutionary events, providing the means to integrate these genomic and geochemical records to establish a well-resolved evolutionary ecology of Chlorobiales. This work demonstrates how phylogenomic analysis can provide an independent source of information for interpreting the lipid biomarker record, and the importance of integrating phylogenomic data with stable carbon isotope analysis in inferring the evolutionary history of microbial metabolisms.

## Supporting information

**S1 Table. Enzymes, NCBI accessions, Genome IDs, alignment, taxonomic diversity, and ML tree model for query sequences.**
(PDF)

**S1 Fig. Maximum likelihood (ML) tree of ATP citrate lyase beta subunit homologs.** (A) midpoint-rooted tree with collapsed clades labeled with taxonomic group names. (B) Higher resolution tree showing crown Chlorobiales and closely related sequences in Nitrospira/ Nitrospinae. Support values indicate approximate likelihood ratio test (aLRT)/ bootstrap (100 replicates). Major clades with bootstrap (BS) support are labeled with respective values. Color bars to the right of the tree indicate autotrophic (green), heterotrophic (orange), or undetermined (white) carbon metabolisms.
(TIF)

**S2 Fig. Protein Structural map of Chlorobium Limicola ATP Citrate Lyase enzyme.** Structural and labeled functional regions regions are colored corresponding to bars above sequence alignments. The amino acid alignment includes are sequences from C.thalassium, C.limicola, Chlorobium sp.445. [23].
(TIF)

**S3 Fig. Bayesian Inference (BI) consensus trees of ATP citrate lyase alpha (A, C) and beta (B, D) subunit homologs.** Trees depicted without (A, B) and with (C, D) Nitrospinae metagenomic sequences from groundwater metagenomic biosampling included in the alignment depicted with arrow. Collapsed clades labeled with taxonomic group names. Support values show consensus posterior probabilities.
(TIF)

# Acknowledgments

We thank Roger Summons for providing his input on project direction and further discussion on results.

# Author Contributions

**Conceptualization:** Madeline M. Paoletti, Gregory P. Fournier.

**Data curation:** Madeline M. Paoletti, Gregory P. Fournier.

**Formal analysis:** Madeline M. Paoletti, Gregory P. Fournier.

**Funding acquisition:** Gregory P. Fournier.

**Investigation:** Madeline M. Paoletti.

**Methodology:** Madeline M. Paoletti, Gregory P. Fournier.

**Project administration:** Gregory P. Fournier.

**Resources:** Gregory P. Fournier.

**Software:** Gregory P. Fournier.

**Supervision:** Gregory P. Fournier.

**Validation:** Madeline M. Paoletti, Gregory P. Fournier.

**Visualization:** Madeline M. Paoletti, Gregory P. Fournier.

**Writing – original draft:** Madeline M. Paoletti.

**Writing – review & editing:** Madeline M. Paoletti, Gregory P. Fournier.

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
