## [Decision Letter · Decision Letter 0]

10 Jun 2022

PONE-D-22-08377Chimeric Inheritance and Crown-Group Acquisitions of Carbon Fixation Genes within ChlorobialesPLOS ONE

Dear Dr. Fournier,

Thank you for submitting your manuscript to PLOS ONE. After careful consideration, we feel that it has merit but does not fully meet PLOS ONE’s publication criteria as it currently stands. Therefore, we invite you to submit a revised version of the manuscript that addresses the points raised during the review process.

We look forward to receiving your revised manuscript.

Kind regards,

Chih-Horng Kuo, Ph.D.

Academic Editor

PLOS ONE

Journal Requirements:

“This work was supported by NSF Integrated Earth Systems award EAR-1615426 to GPF, and the Dept. of Biological Sciences at Wellesley College Senior Honors Thesis Research Program.”

“This work was supported by the National Science Foundation Integrated Earth Systems award EAR grant no. 1615426 to GPF (https://www.nsf.gov/pubs/2016/nsf16589/nsf16589.htm).

3. Please ensure that you refer to Figures 5 and 6 in your text as, if accepted, production will need this reference to link the reader to the figure.

Reviewers' comments:

Reviewer's Responses to Questions

**Comments to the Author**

1. Is the manuscript technically sound, and do the data support the conclusions?

Reviewer #1: Partly

Reviewer #2: Yes

2. Has the statistical analysis been performed appropriately and rigorously? 

Reviewer #1: N/A

Reviewer #2: Yes

3. Have the authors made all data underlying the findings in their manuscript fully available?

Reviewer #1: Yes

Reviewer #2: Yes

4. Is the manuscript presented in an intelligible fashion and written in standard English?

Reviewer #1: Yes

Reviewer #2: Yes

5. Review Comments to the Author

Reviewer #1: The manuscript of Paoletti and Fournier explores the phylogenetic history of the reverse TCA cycle in the order Chlorobiales. Biomarkers, mostly derived from aromatic carotenoids once believed to be specific markers for phototrophiic sulfur bacteria, have been oberved in the rock record extending back about 1.64 Ga. However, recent studies have shown that aromatic carotenoids are also produced by many modern-day cyanobacteria. To properly interpret isotopic fractionation of carbon in biomarkers requires knowledge of the carbon fixation pathways in use by organisms that produced the biomarkers. Thus, this is a timely study that will be of interest to a wide audience. Comments for the authors follow.

1. Lines 31 to 33. Considering that renieratene and isorenieratene have recently been shown to be produced by cyanobacteria that can also produce synechoxanthin, okenone may be the only reliable biomarker associated with (purple) sulfur bacteria. Although chlorobactene has not been shown to be produced by cyanobacteria yet, any cyanobacterium that can synthesize isorenieratene could in principle make chlorobactene, assuming the carotene desaturase could function with gamma-carotene as substrate.

2. Lines 50-51. The CBB cycle also occurs in many chemolithoautotrophic bacteria, not just photoautotrophs.

3. Lines 71-73. I am not sure of the “official” state of taxonomy concerning the order Chlorobiaceae, but I think there are only three genera (see Imhoff 2003) now: Prosthecochloris, Chlorobium and Chlorobaculum. Pelodictyon (note spelling) and Ancalochloris are no longer considered to be valid taxa. The status of Chloroherpeton is less clear, but it is probably the type genus of a separate family, because Chloroherpeton are metabolically distinct from all other green sulfur bacteria in that they can only oxidize sulfide to the level of sulfur/polysulfide.

4. Lines 73-75. The authors should probably cite Liu et al., 2012, Front. Microbiol. 3, 185. Ignavibacterium album is an aerobe that can grow as a facultative or fermentative anaerobe as well. It has a full aerobic electron transfer chain and has been demonstrated to grow by aerobic respiration.

5. Lines 77 to 79. Thermochlorobacter spp. has been cultured but not axenically, only in enrichment cultures (Tank et al. 2017). It is clearly a full aerobe with a complete electron transfer chain to match, with metabolic properties similar to those of Chloroacidobacterium. Thermochlorobacter and related taxa are unable to oxidize sulfide at all, do not fix carbon dioxide, but have a photosynthetic apparatus similar to that of GSB. These bacteria are not “sulfur bacteria” at all, unless assimilatory sulfate reduction is included in the definition, but then just about all bacteria would be sulfur bacteria.

6. Lines 71 to 81. The authors do not present any analysis of the current state of the 16S rRNA phylogeny of GSB, nor do they present any phylogenetic trees based upon shared proteins among these organisms. This makes the interpretation of the trees presented for the rTCA cycle enzymes difficult for those less familiar with the taxonomy of green bacteria. A 16S rRNA tree finally appears in Figure 11.

7. The importance of the last several points is that the earliest diverging members of the Chlorobiales and the sister taxon Ignavibacteria is that these organisms were aerobes, not anaerobes as once generally believed. This has important implications for how the authors interpret their trees and the information derived from them. Even if the authors do not agree with these data, they must present the context for their intepretation, so I would suggest that the authors begin with a section describing protein and 16S rRNA trees to establish the relationships of the organisms under consideration with other bacteria.

8. Lines 109-111. Again, considering that I. album and Thermochlorobacter are aerobes, and that all GSB have retained the genes for at least one terminal cytochrome/quinol oxidase, it seems likely that the GSB are derived from aerobic ancestors that gained phototrophy and many other traits based on HGT.

9. Figure 3. Chloracidobacterium thermophilum is not an autotroph. It is a microaerophilic heterotroph that mostly consumes amino acids, especially branched chain amino acids. It likely uses the partial reverse TCA cycle to carboxylate succinyl-CoA to make 2-oxoglutarate.

10. Line 181. Not exactly. Thermochlorobacter doesn’t have fumarate reductase because it has an oxidative TCA cycle, not a reductive one.

11. Figure 5B. This figure largely replicates what I expect a 16S rRNA tree would look like, and probably a tree based on concatenated core proteins as well. In panel A, I am not sure whether there are any autotrophic Acidobacteria, but I don’t think there are any.

12. Lines 206-210. Organisms that can degrade branched chain amino acids as carbon and nitrogen sources can do so by making succinate/succinyl-CoA. This can be carboxylated to produce 2-oxoglutarate by the partial reverse TCA cycle to supply precursors for Chl/BChl biosynthesis. This is very beneficial to organisms that are unable to grow as full autotrophs.

13. Figure 6A. Not all Chlorobiales are autotrophs (see panel B below).

14. Line 218 to 225. Presumably this refers to Figure 6, but panel 6B looks like a 16S rRNA tree to me with early diverging Chloroherpeton and Thermochlorobacter sequences. What is misrooted about this?

15. Figure 9A. Chlorobium sp. 445 should be orange, not green (see panel B).

16. Lines 305 to 307. Seems that the ancestral state was an oxidative rather than a reductive TCA cycle, so gain more likely than loss?

17. Lines 330 to 332. Is pyruvate dehydrogenase present in the aerobic heterotrophs?

18. Figure 11. This is an interesting figure, but it could perhaps be made still more informative by including some other events as well. For example, all the organisms in the gray shading acquired the genes for type-1 reaction centers, FMO bacteriochlorophyll a binding protein, and the genes for bacteriochlorophyll c biosynthesis and chlorosome assembly. The Chlorobiaceae gained the ability to oxidize sulfur/polysulfide to sulfate, and Chlorobaculum gained the ability to oxidize thiosulfate to sulfate.

19. Most (but not all) members of the Chlorobiaceae can synthesize chlorobactene or isorenieratene; a few seem to have lost CrtQ. Chloroherpeton and Thermochlorobacter cannot make aromatic carotenoids, and that seems to be trait that was acquired relatively late by GSB. However, exactly when this trait was acquired is not clear. All strains in the shaded area, as well as I. album, can synthesisize carotenoids, which are essential for organisms that use chlorophylls to perform phototrophy in the presence of oxygen.

Reviewer #2: This work presents an extensive phylogenetic study of all enzymes involved in the rTCA cycle of Chlorobiales and closely related species. the work is framed within the context of the origin of autotrophy within this group and its potential implications for the use of the ~1.64 biomarkers traditionally used as a calibration point for molecular clock analyses.

From a phylogenetic perspective, the work is well done and comprehensive. I am not aware of other studies that have analyzed all the enzymes in this pathway and, therefore, it adds an important piece to reconstruct its evolutionary history. The methodology used to infer the phylogenies is standard and appropriate. However, it would have been useful to see a discussion on the accuracy of all the phylogenies used as basis for the conclusions drawn. Prokaryote phylogenies are notoriously sensitive to many factors, from alignment length to species composition. Most bootstrap values, especially those of the chlorobiales/sister group node, are relatively high, which suggest these trees can be used with a decent amount of confidence. However, I think it is risky to draw such wide-ranging conclusions without even discussing potential pitfall in phylogenetic reconstruction.

My major concern, that I think can be addressed easily with some clarifications, is the connection of the history of rTCA and autotrophy with the validity of the 1.64 Ga biomarker. Lines 38-40 of the introduction state that the use of these biomarkers as a calibration for dating Chlorobiales is “into question” but I was not able to see how the results presented in this work may or may not support the use of these biomarkers. This is extremely important because these biomarkers are among the very few available for prokaryote molecular clock analyses and, therefore, any alteration of their use would have profound impacts on the field.

Overall, I find that this work adds interesting information on the rTCA cycle evolution (provided a discussion on phylogenetic accuracy is added, as mentioned above) but its framing for its relevance to the timing of chlorobiales to be unclear. I would encourage the authors to either reconsider this framework, since the work would stand on its own without needing the molecular clock angle, or clarify its implication for the use of the 1.64 Ga biomarkers.

6. PLOS authors have the option to publish the peer review history of their article (what does this mean?). If published, this will include your full peer review and any attached files.

Reviewer #1: No

Reviewer #2: No

---

## [Author Response · Author response to Decision Letter 0]

1 Aug 2022

Reviewer's Responses to Questions

Comments to the Author

1. Is the manuscript technically sound, and do the data support the conclusions?

Reviewer #1: Partly

Reviewer #2: Yes

2. Has the statistical analysis been performed appropriately and rigorously?

Reviewer #1: N/A

Reviewer #2: Yes

3. Have the authors made all data underlying the findings in their manuscript fully available?

Reviewer #1: Yes

Reviewer #2: Yes

4. Is the manuscript presented in an intelligible fashion and written in standard English?

PLOS ONE does not copy edit accepted manuscripts, so the language in submitted articles must be clear, correct, and unambiguous. Any typographical or grammatical errors should be corrected at revision, so please note any specific errors here.

Reviewer #1: Yes

Reviewer #2: Yes

5. Review Comments to the Author

Reviewer #1: The manuscript of Paoletti and Fournier explores the phylogenetic history of the reverse TCA cycle in the order Chlorobiales. Biomarkers, mostly derived from aromatic carotenoids once believed to be specific markers for phototrophiic sulfur bacteria, have been observed in the rock record extending back about 1.64 Ga. However, recent studies have shown that aromatic carotenoids are also produced by many modern-day cyanobacteria. To properly interpret isotopic fractionation of carbon in biomarkers requires knowledge of the carbon fixation pathways in use by organisms that produced the biomarkers. Thus, this is a timely study that will be of interest to a wide audience. Comments for the authors follow.

1. Lines 31 to 33. Considering that renieratene and isorenieratene have recently been shown to be produced by cyanobacteria that can also produce synechoxanthin, okenone may be the only reliable biomarker associated with (purple) sulfur bacteria. Although chlorobactene has not been shown to be produced by cyanobacteria yet, any cyanobacterium that can synthesize isorenieratene could in principle make chlorobactene, assuming the carotene desaturase could function with gamma-carotene as substrate.

We thank the reviewers for pointing out this detail, and now expand our discussion of these biomarkers in Cyanobacteria to mention the possibility of ancient cyanobacterial sources of chlorobactene. We have added the text: 

“Carotenoid biomarkers within Neoproterozoic marine sediments (1000-542 Ma) have been shown to be primarily comprised of cyanobacterial renierapurpurane, with small amounts of the Chlorobiaceae-associated biomarker isorenieratane. Furthermore, cyanobacteria producing synechoxanthin have been shown to be capable of producing both isorenieratene and, in principle, chlorobactene, given their enzymatic repertoire (cite?). This brings the evidence for GSB at 1.64 Ga BCF formations into question, as well as the practice of using these biomarkers as a calibration for dating Chlorobiales in molecular clock analyses, since cyanobacteria could potentially be the source of the carotenoid derivatives found in these deposits12,13” in lines 16-24 of page 2.

2. Lines 50-51. The CBB cycle also occurs in many chemolithoautotrophic bacteria, not just photoautotrophs.

The reviewer is indeed correct, we now clarify this detail:

Page 2, line: 36-39: “The CBB cycle is found in both photoautotrophic bacteria such as cyanobacteria and purple sulfur bacteria, as well as some chemolithoauthotrophs; it fixes three molecules of CO2 for the synthesis of one 3-phoshoglycerate molecule using ribulose-1,5-bisphosphate carboxylase/oxygenase (RuBisCo).”

3. Lines 71-73. I am not sure of the “official” state of taxonomy concerning the order Chlorobiaceae, but I think there are only three genera (see Imhoff 2003) now: Prosthecochloris, Chlorobium and Chlorobaculum. Pelodictyon (note spelling) and Ancalochloris are no longer considered to be valid taxa. The status of Chloroherpeton is less clear, but it is probably the type genus of a separate family, because Chloroherpeton are metabolically distinct from all other green sulfur bacteria in that they can only oxidize sulfide to the level of sulfur/polysulfide.

The reviewer is correct. The taxonomic designations used in our original manuscript were based on (Imhoff, 2003). The author of this work has since published an updated classification schema for GSB solely using the genera Chlorobium, Chlorobaculum, Prosthecochloris, and Chloroherpeton. The change is noted in our reference #30 on page 17 and in the manuscript page 3, lines 63-65.

There has been some recent debate due from new isolates, particularly from environmental metagenomic sequences, grouping within the more metabolically distinct Chloropeton genera for further revisions to this taxonomic labeling, such as creating distinct families of Chloroherpetonaceae and Thermochlorobacteriacea (Liu et. al., 2012). However, since these labels have not been adopted in the larger field yet and much debate remains on the physiologies of characterized photoheterotrophs in the order Chlorobiales, we retain the taxonomic labels suggested by Imhoff. The taxonomic labels in our datasets and resulting phylogenies reflect those in use by NCBI at the time of the analysis, as these are automatically populated by our scripts. 

4. Lines 73-75. The authors should probably cite Liu et al., 2012, Front. Microbiol. 3, 185. Ignavibacterium album is an aerobe that can grow as a facultative or fermentative anaerobe as well. It has a full aerobic electron transfer chain and has been demonstrated to grow by aerobic respiration.

The reviewer is correct–however, Ignavibacteriales, while the sister clade to Chlorobiales, is still quite evolutionary distant, and the relevance of their extant energy metabolisms to the ancestral state of autotrophy vs. heterotrophy in Chlorobiales is not immediately evident. Making such a connection would require a significant ancestral trait reconstruction effort, which is beyond the scope of the work presented here, and does not directly bear on the evolutionary history of the much more derived groups within Chlorobiales that we investigate. 

5. Lines 77 to 79. Thermochlorobacter spp. has been cultured but not axenically, only in enrichment cultures (Tank et al. 2017). It is clearly a full aerobe with a complete electron transfer chain to match, with metabolic properties similar to those of Chloroacidobacterium. Thermochlorobacter and related taxa are unable to oxidize sulfide at all, do not fix carbon dioxide, but have a photosynthetic apparatus similar to that of GSB. These bacteria are not “sulfur bacteria” at all, unless assimilatory sulfate reduction is included in the definition, but then just about all bacteria would be sulfur bacteria.

The reviewer’s comment points out the importance in clarifying the distinction between “Green sulfur bacteria” as a phenotypic descriptor, vs. its use as an interchangeable clade name for “Chlorobiales”. In order to avoid the possible confusion between these two usages, we now only use the order name “Chlorobiales” to refer to the members of this group including the photoheterotroph clade and the family name “Chlorobiaceae” when referring to traditional Chlorobi “green sulfur bacteria” . We still use “GSB” in the introduction when referring to previously published work.

6. Lines 71 to 81. The authors do not present any analysis of the current state of the 16S rRNA phylogeny of GSB, nor do they present any phylogenetic trees based upon shared proteins among these organisms. This makes the interpretation of the trees presented for the rTCA cycle enzymes difficult for those less familiar with the taxonomy of green bacteria. A 16S rRNA tree finally appears in Figure 11.

Our understanding is that the evolutionary relationships between these groups are well-established, and agree with our own 16S tree presented here. We agree with the reviewer, that clearly communicating this species tree early in the manuscript will aid the reader in interpreting the gene trees. In the introduction, we now include a brief description of the current phylogeny of GSB, and a figure (lines 80-84) showing these relationships on pg.3 , lines 68-71 & 76-79-113: 

“With the exception of the unclassified thermophiles, Chlorobiales are strictly anaerobic, obligate photoautotrophs that use the rTCA cycle. Within this group, Chlorobium, Chlorobaculum, and Prosthecochloris genera form a clade that synthesizes aromatic carotenoid compounds…This distribution of physiological traits in Chlorobiales may be explained by vertical inheritance and loss, or by HGT, depending on the inferred phylogenies of the individual gene families providing these functions.”

Fig 2. 16S rRNA tree of Chlorobiales and Ignavibacterium (outgroup) showing taxonomic distributions of a selection of traits. Labeled brackets indicate the taxonomic distribution of phototrophy and autotrophy within this group. The green box indicates groups known to synthesize aromatic carotenoid lipid biomarkers.

7. The importance of the last several points is that the earliest diverging members of the Chlorobiales and the sister taxon Ignavibacteria is that these organisms were aerobes, not anaerobes as once generally believed. This has important implications for how the authors interpret their trees and the information derived from them. Even if the authors do not agree with these data, they must present the context for their interpretation, so I would suggest that the authors begin with a section describing protein and 16S rRNA trees to establish the relationships of the organisms under consideration with other bacteria.

We now begin with such a section that clearly provides the evolutionary framework for these lineages.

8. Lines 109-111. Again, considering that I. album and Thermochlorobacter are aerobes, and that all GSB have retained the genes for at least one terminal cytochrome/quinol oxidase, it seems likely that the GSB are derived from aerobic ancestors that gained phototrophy and many other traits based on HGT.

The reviewer proposes an interesting evolutionary scenario that would reconstruct aerobic respiration as the ancestral metabolism for the Ignavibacteriales/Chlorobiales clade. While this is certainly possible, our work here is concerned with the evolution of carbon fixation within phototrophic Chlorobiales (an apomorphy-defined clade including extant Chlorobiales and their phototrophic ancestors). Reconstructing earlier evolutionary stages (pre-phototrophy) is an interesting and complex question that would require ancestral reconstruction of many gene families and traits, that is beyond the scope of the work presented here, and does not bear directly on the question of photoheterotrophy vs. photoautotrophy in ancient Chlorobiales. In other words, an early ancestral aerobic metabolism within stem Chlorobiales is not an “alternative” hypothesis to be presented, but an independent hypothesis that does not directly bear on subsequent evolutionary scenarios for carbon fixation that may have occurred within phototrophic Chlorobiales, and so does not aid in discriminating between them.

9. Figure 3. Chloracidobacterium thermophilum is not an autotroph. It is a microaerophilic heterotroph that mostly consumes amino acids, especially branched chain amino acids. It likely uses the partial reverse TCA cycle to carboxylate succinyl-CoA to make 2-oxoglutarate. 

We thank the reviewer for pointing this out, and have updated Figure 3 to reflect this, as well as other figures where Acidobacteria were incorrectly labeled as autotrophs (Fig 2, Fig 3, Fig 5, Fig 7, Fig 10). All members of this phyla that have been cultured are heterotrophs (Kielak et. al., 2016).

10. Line 181. Not exactly. Thermochlorobacter doesn’t have fumarate reductase because it has an oxidative TCA cycle, not a reductive one: 

The reviewer is correct in their interpretation. We have now expanded this section to specifically point out that the gene encoding fumarate reductase was apparently lost in the ancestor lineage of Thermochlorobacter and related photoheterotrophs on pg 7 lines 202-205:

“The ML tree of fumarate reductase homologs recovers the monophyly of Chlorobiales with the exception of the thermophilic photoheterotrophs Thermochlorobacter, and Chlorobium sp.445. Observed sequences are closely related to those from Candidatus Lambdaproteobacteria, Geobacter, and Desulfuromonadales bacterium (Fig 5).”

11. Figure 5B. This figure largely replicates what I expect a 16S rRNA tree would look like, and probably a tree based on concatenated core proteins as well. In panel A, I am not sure whether there are any autotrophic Acidobacteria, but I don’t think there are any.

This error has been noted and fixed as stated in comment #9. 

12. Lines 206-210. Organisms that can degrade branched chain amino acids as carbon and nitrogen sources can do so by making succinate/succinyl-CoA. This can be carboxylated to produce 2-oxoglutarate by the partial reverse TCA cycle to supply precursors for Chl/BChl biosynthesis. This is very beneficial to organisms that are unable to grow as full autotrophs.

We thank the reviewer for this interesting detail. We now include a brief comment noting this possible connection on pg.7, lines 233-236: “In bacteriochlorophyll biosynthesis, succinate/succinyl-CoA can be carboxylated by a partial rTCA cyle to create 2-oxoglutate as precursor molecules, providing a pre-adaptation in the form of a selective advantage to photoheterotrophic lineages34,73.”

13. Figure 6A. Not all Chlorobiales are autotrophs (see panel B below).

The error in the figure labeling the Chlorobiales collapsed branch as all autotrophs has been noted and changed in panel A to correctly depict it a mixed grouping. 

14. Line 218 to 225. Presumably this refers to Figure 6, but panel 6B looks like a 16S rRNA tree to me with early diverging Chloroherpeton and Thermochlorobacter sequences. What is misrooted about this?

The tree is described as misrooted because, in contrast to the 16S tree and other published phylogenies, the photoheterotrophs do not group with Chloroherpeton, but are the most deeply diverging lineage. The expected root would place Chloroherpeton together with the photoheterotrophs as a clade, with other members of Chlorobiales as the sibling group. We believe this is already clearly stated in the text on pg 8 lines 263-265.

“The ML tree does not recover the sibling grouping of C.thalassium and thermophilic photoheterotrophic lineages, suggesting a possible misrooting of Chlorobiales in this tree.”

15. Figure 9A. Chlorobium sp. 445 should be orange, not green (see panel B).

Noted and updated in figure. 

16. Lines 305 to 307. Seems that the ancestral state was an oxidative rather than a reductive TCA cycle, so gain more likely than loss?

In this particular case, the gene in question (ATP citrate lyase) has experienced deletions potentially indicative of a loss of function. Since other photoheterotrophs lack this gene entirely, we infer this to be a remnant that points to ancestral presence of more rTCA genes within this lineage. The remaining TCA genes in this group are now part of a partial pathway that is likely oxidative. This C. sp. 445 sequence groups basally with other Chlorobiales, consistent with a shared common ancestry and thus loss in other photoheterotropic lineages, rather than an independent gain, although more complex histories are certainly possible. We now qualify our observations by mentioning this scenario:

Lines 339-342, Page 10: “If not mis-rooted, this tree topology could also be explained by an ancestral loss of ATP citrate lyase in the photoheterotroph ancestor, followed by a secondary acquisition in Chlorobium sp. 445, either from the Nitrospirae/Nitrospinae lineage, or an HGT donor group common to both, although this is a less parsimonious scenario.”

17. Lines 330 to 332. Is pyruvate dehydrogenase present in the aerobic heterotrophs?

As far as our results show, pyruvate dehydrogenase is not present in the Chlorobiales heterotrophs, which we believe is demonstrated by our tree Fig 11 and now made clearer in the text pg. 10, lines 373-374: “This enzyme is one of only three that are present in the rTCA cycle and not the oxidative TCA cycle24 and was apparently lost within the photoheterotrophic Chlorobiaceae lineages, as they do not appear with other Chlorobiales in the gene tree (Fig 10B).”

18. Figure 11. This is an interesting figure, but it could perhaps be made still more informative by including some other events as well. For example, all the organisms in the gray shading acquired the genes for type-1 reaction centers, FMO bacteriochlorophyll a binding protein, and the genes for bacteriochlorophyll c biosynthesis and chlorosome assembly. The Chlorobiaceae gained the ability to oxidize sulfur/polysulfide to sulfate, and Chlorobaculum gained the ability to oxidize thiosulfate to sulfate.

The acquisition of these additional traits are indeed important and interesting parts of the evolution of Chlorobiales. In fact, our figure already denotes the acquisition of type-1 reaction centers (PS1), and the acquisition of bacteriochlorophyll synthesis and binding proteins is implied by the “Photoheterotrophy” label. We agree that “chlorosomes” is a valuable addition to this figure, as the role of chlorosomes is specifically discussed in the manuscript. We now see that “PS1” is insufficiently descriptive. Therefore, we have updated this figure to more precisely and less redundantly describe the crown Chlorobiales box as containing “type-1 reaction centers, bacteriochlorophyll, chlorosomes”. Other physiological traits relating to sulfur oxidation are less directly connected to the results and narrative of the paper, and we have opted to not include these, to keep the figure as simple as readable as possible. 

19. Most (but not all) members of the Chlorobiaceae can synthesize chlorobactene or isorenieratene; a few seem to have lost CrtQ. Chloroherpeton and Thermochlorobacter cannot make aromatic carotenoids, and that seems to be trait that was acquired relatively late by GSB. However, exactly when this trait was acquired is not clear. All strains in the shaded area, as well as I. album, can synthesisize carotenoids, which are essential for organisms that use chlorophylls to perform phototrophy in the presence of oxygen.

We agree, our current description as “carotenoid pigments” was insufficiently precise. We agree, our current description as “carotenoid pigments” was insufficiently precise. We have removed this label in the discussion figure (Fig. 12). We now include this correctly labeled group 

"aromatic carotenoid biosynthesis" in the newly added overview Figure 2. We have also added a section to the discussion to clarify how the taxonomic distribution of biomarker synthesis impacts these hypotheses: 

Lines 490-495 Page 13: “Furthermore, aromatic carotenoid synthesis is absent within Chloroherpeton and photoheterotroph lineages, suggesting that the most parsimonious history of these genes would have them acquired after the crown divergence of Chlorobiaceae. This argues against the BCF biomarkers being sourced from stem group Chlorobiales, unless there were subsequent losses of biomarker synthesis genes within the crown group.”

Reviewer #2: This work presents an extensive phylogenetic study of all enzymes involved in the rTCA cycle of Chlorobiales and closely related species. the work is framed within the context of the origin of autotrophy within this group and its potential implications for the use of the ~1.64 biomarkers traditionally used as a calibration point for molecular clock analyses.

From a phylogenetic perspective, the work is well done and comprehensive. I am not aware of other studies that have analyzed all the enzymes in this pathway and, therefore, it adds an important piece to reconstruct its evolutionary history. The methodology used to infer the phylogenies is standard and appropriate. However, it would have been useful to see a discussion on the accuracy of all the phylogenies used as basis for the conclusions drawn. Prokaryote phylogenies are notoriously sensitive to many factors, from alignment length to species composition. Most bootstrap values, especially those of the chlorobiales/sister group node, are relatively high, which suggest these trees can be used with a decent amount of confidence. However, I think it is risky to draw such wide-ranging conclusions without even discussing potential pitfall in phylogenetic reconstruction.

The reviewer is correct in that tree inferences are often limited by the accuracy of phylogenetic reconstruction, as impacted by the available sequence data and models we have. These pitfalls relate to interpreting topology, when alternative topologies not recovered by the phylogeny may also be potentially correct, leading to different conclusions. In the work presented here, for many of these gene families, the two hypotheses being compared are (1) shared common ancestry and vertical inheritance of the gene within crown group Chlorobiales; vs. (2) independent acquisition via HGT by different sub-groups within Chlorobiales, and no shared common ancestry within crown group Chlorobiales. In the cases where we propose (2), the homologs found within different groups of Chlorobiales are so distantly related that it is extremely unlikely that the observed polyphyly is a result of phylogenetic uncertainty or reconstruction error. In other words, even with the limitations of phylogenetic reconstruction, these gene histories adequately discriminate between these two hypotheses. However, the inferred HGT donor groups in these cases may be impacted by the kinds of uncertainty mentioned by the reviewer. Therefore, we now clarify that both taxon sampling and phylogenetic reconstruction limitations may impact the exact placement of these groups, limiting our ability to conclusively identify HGT donors or trace the deeper history of the acquisitions of these gene families.

Page 11, lines 415-418: “While current taxon sampling and the inherent limitations of phylogenetic inference prevent, in most cases, the identification of specific HGT donor groups for these genes, their histories are unambiguously incompatible with the alternative hypothesis of vertical inheritance from stem Chlorobiales.”

My major concern, that I think can be addressed easily with some clarifications, is the connection of the history of rTCA and autotrophy with the validity of the 1.64 Ga biomarker. Lines 38-40 of the introduction state that the use of these biomarkers as a calibration for dating Chlorobiales is “into question” but I was not able to see how the results presented in this work may or may not support the use of these biomarkers. This is extremely important because these biomarkers are among the very few available for prokaryote molecular clock analyses and, therefore, any alteration of their use would have profound impacts on the field.

Thank you for pointing out that additional clarity is needed here. We expanded our conclusions to now more explicitly make predictions and emphasize the significance of our findings with respect to interpreting the carotenoid biomarker record:

Lines 544-552, Page 14: “This specifically implies that a “GSB-like” carbon isotopic fractionation within preserved aromatic carotenoid biomarkers should not be the standard of evidence to infer Chlorobiales as their biological source. Rather, we would expect that older carotenoid material, such as that obtained from the BCF, should instead show fractionations consistent with heterotrophy, closer to that of bulk organic carbon within the system. This is especially important given the proposed alternative cyanobacterial origin for these lipids, which may be falsely inferred in the absence of observing the expected “rTCA” signature fractionation. “

Overall, I find that this work adds interesting information on the rTCA cycle evolution (provided a discussion on phylogenetic accuracy is added, as mentioned above) but its framing for its relevance to the timing of chlorobiales to be unclear. I would encourage the authors to either reconsider this framework, since the work would stand on its own without needing the molecular clock angle, or clarify its implication for the use of the 1.64 Ga biomarkers.

6. PLOS authors have the option to publish the peer review history of their article (what does this mean?). If published, this will include your full peer review and any attached files.

Do you want your identity to be public for this peer review? For information about this choice, including consent withdrawal, please see our Privacy Policy.

Reviewer #1: No

Reviewer #2: No

---

## [Decision Letter · Decision Letter 1]

9 Sep 2022

PONE-D-22-08377R1Chimeric Inheritance and Crown-Group Acquisitions of Carbon Fixation Genes within Chlorobiales: Origins of autotrophy in Chlorobiales and implication for geological biomarkersPLOS ONE

Dear Dr. Fournier,

Thank you for submitting your manuscript to PLOS ONE. After careful consideration, we feel that it has merit but does not fully meet PLOS ONE’s publication criteria as it currently stands. Therefore, we invite you to submit a revised version of the manuscript that addresses the points raised during the review process.

We look forward to receiving your revised manuscript.

Kind regards,

Chih-Horng Kuo, Ph.D.

Academic Editor

PLOS ONE

Journal Requirements:

Additional Editor Comments (if provided):

Congratulations on the nice work. Please consider the minor suggestion made by Reviewer 1, then I am ready to recommend formal acceptance.

Reviewers' comments:

Reviewer's Responses to Questions

**Comments to the Author**

1. If the authors have adequately addressed your comments raised in a previous round of review and you feel that this manuscript is now acceptable for publication, you may indicate that here to bypass the “Comments to the Author” section, enter your conflict of interest statement in the “Confidential to Editor” section, and submit your "Accept" recommendation.

Reviewer #1: (No Response)

Reviewer #2: All comments have been addressed

2. Is the manuscript technically sound, and do the data support the conclusions?

Reviewer #1: Yes

Reviewer #2: Yes

3. Has the statistical analysis been performed appropriately and rigorously? 

Reviewer #1: N/A

Reviewer #2: Yes

4. Have the authors made all data underlying the findings in their manuscript fully available?

Reviewer #1: Yes

Reviewer #2: Yes

5. Is the manuscript presented in an intelligible fashion and written in standard English?

Reviewer #1: Yes

Reviewer #2: Yes

6. Review Comments to the Author

Reviewer #1: The authors have done an excellent job of addressing my extensive comments, with only a single exception. I don't think Chloroherpeton belongs i the Chlorobiaceae. It should be a 2nd family, Chloroherpetonaceae, in the Chlorobiales. Other than this, I am satisfied with the modifications made to the original manuscript, which I believe has been improved.

Reviewer #2: The authors responded thoroughly and satisfactorily to the comments made in the previous round of review.

7. PLOS authors have the option to publish the peer review history of their article (what does this mean?). If published, this will include your full peer review and any attached files.

Reviewer #1: No

Reviewer #2: No

---

## [Author Response · Author response to Decision Letter 1]

14 Sep 2022

Reviewer #1 Comment: The authors have done an excellent job of addressing my extensive

comments, with only a single exception. I don't think Chloroherpeton belongs in the Chlorobiaceae. It

should be a 2nd family, Chloroherpetonaceae, in the Chlorobiales. Other than this, I am satisfied with the

modifications made to the original manuscript, which I believe has been improved.

Response: We thank reviewer one for this point. After considering newer literature on the classification system for

Chlorobi along with NCBI taxonomic identifications, we have decided to place Chloroherpeton in the

family Chloroherpetonaceae, separate from Chlorobiaceae. Besides altering taxonomic names in the

manuscript where required and relabeling Figure 12, we believe this change does not further alter our

findings or conclusions.

---

## [Editor Report · Decision Letter 2]

19 Sep 2022

Chimeric Inheritance and Crown-Group Acquisitions of Carbon Fixation Genes within Chlorobiales: Origins of autotrophy in Chlorobiales and implication for geological biomarkers

PONE-D-22-08377R2

Dear Dr. Fournier,

We’re pleased to inform you that your manuscript has been judged scientifically suitable for publication and will be formally accepted for publication once it meets all outstanding technical requirements.

Kind regards,

Chih-Horng Kuo, Ph.D.

Academic Editor

PLOS ONE
---

## [Editor Report · Acceptance letter]

22 Sep 2022

PONE-D-22-08377R2 

Chimeric inheritance and crown-group acquisitions of carbon fixation genes within Chlorobiales: Origins of autotrophy in Chlorobiales and implication for geological biomarkers 

Dear Dr. Fournier:

I'm pleased to inform you that your manuscript has been deemed suitable for publication in PLOS ONE. Congratulations! Your manuscript is now with our production department. 

Kind regards, 

on behalf of

Dr. Chih-Horng Kuo 

Academic Editor

PLOS ONE